# Retrieval practice facilitates memory updating by enhancing and differentiating medial prefrontal cortex representations

**Zhifang Ye[1], Liang Shi[1], Anqi Li[1], Chuansheng Chen[2], Gui Xue[1]\***

[1]State Key Laboratory of Cognitive Neuroscience and Learning & IDG/McGovern Institute of Brain Research, Beijing Normal University, Beijing, China; [2]Department of Psychological Science, University of California, Irvine, Irvine, United States

**Abstract** Updating old memories with new, more current information is critical for human survival, yet the neural mechanisms for memory updating in general and the effect of retrieval practice in particular are poorly understood. Using a three-day A-B/A-C memory updating paradigm, we found that compared to restudy, retrieval practice could strengthen new A-C memories and reduce old A-B memory intrusion, but did not suppress A-B memories. Neural activation pattern analysis revealed that compared to restudy, retrieval practice led to stronger target representation in the medial prefrontal cortex (MPFC) during the final test. Critically, it was only under the retrieval practice condition that the MPFC showed strong and comparable competitor evidence for both correct and incorrect trials during final test, and that the MPFC target representation during updating was predictive of subsequent memory. These results suggest that retrieval practice is able to facilitate memory updating by strongly engaging MPFC mechanisms in memory integration, differentiation and consolidation.

**\*For correspondence:**
gxue@bnu.edu.cn

## Introduction

Being able to remember and retain the most current information in a dynamically changing world requires the capacity to update one's memory in a goal-directed manner. Updating occurs when some information is downgraded as outdated or irrelevant, and newer information is promoted as its replacement. Examples range from something as simple as trying to remember one's new home address and phone number, to replacing maladaptive memories with adaptive ones in a therapeutic setting. Successful updating of memory involves strengthening the more current memory trace, weakening the older information, and/or differentiating old and new memories, in order to ensure that the old memory is less interfering.

Ample behavioral evidence suggests that although memory updating can be fostered by repeatedly studying the new replacement information, memory is more successfully updated by the act of retrieving the new knowledge via self-tests, a process called retrieval practice (*Roediger and Butler, 2011*). Compared to simple restudy of the same material again, retrieval of learnt information leads not only to better retention of relevant memory when no obvious interference is involved (*Karpicke and Roediger, 2008*; *Pyc and Rawson, 2010*), but also to better inhibition of competing, outdated memories (*Anderson et al., 1994*), reduced proactive interference (*Szpunar et al., 2008*), enhanced memory integration (*Hupbach et al., 2007*), and greater susceptibility to subsequent modification (*Chan et al., 2009*). These findings indicate that retrieval practice modifies the state of memory according to mnemonic goals. Despite this ubiquitous behavioral effect, the neural basis for memory updating and the mechanism that underlies the retrieval-practice benefit remain unclear.

A large body of work on memory reconsolidation (*Dudai, 2006*) suggests that consolidated memories, when retrieved and reactivated, enter into a transient, labile state, rendering them vulnerable

to modification (*Lee et al., 2017*). Electrical shock (*Kroes et al., 2014*) and pharmacological treatments (*Kindt et al., 2009*; *Nader et al., 2000*) that block protein synthesis can cause long-lasting impairment of existing memories when they are reactivated. Reactivated memories can also be disrupted by behavioral methods, including retrieval-extinction (*Schiller et al., 2010*; *Xue et al., 2012*), counterconditioning (*Goltseker et al., 2017*), and interference approaches (*James et al., 2015*), which have been found to reduce the expression of fear or drug memories and amygdala responses associated with fear (*Agren et al., 2012*). Using representational similarity analysis, recent studies further show that the reactivated memories could be selectively strengthened (*Jonker et al., 2018*; *Lu et al., 2015*; *Xue et al., 2010*), weakened (*Wimber et al., 2015*), integrated (*Schlichting et al., 2015*), and/or differentiated (*Hulbert and Norman, 2015*), depending on the mnemonic goals and the characteristics of the reactivated memory (*Tambini and Davachi, 2019*).

Emerging studies suggest that the medial prefrontal cortex (MPFC) probably plays an important role in the rapid formation of cortical memories, especially during retrieval practice. Specifically, it has been hypothesized that retrieval practice will reactivate related memory traces, and that the MPFC is able to develop integrated neocortical representations of these memory traces rapidly, in a way that resembles the characteristics of rapid system consolidation (*Antony et al., 2017*). Consistently, the MPFC has been found to be involved in the integration and updating of reactivated memory traces (*Gilboa and Marlatte, 2017*; *Preston and Eichenbaum, 2013*). For example, the MPFC is critically involved in encoding novel but related information into existing knowledge (*Sommer, 2016*), in representing overlapping memories (*Tompary and Davachi, 2017*), and in inferring relationships between distinct events that share common features (*Zeithamova et al., 2012*). In these studies, the reactivation of related memories has been found to be important for the MPFC-mediated processes. Nevertheless, it is unclear how the MPFC would be involved when competing memories were reactivated.

Several studies suggest that the lateral prefrontal cortex (LPFC) may play a role in regulating reactivated memories to support memory changes. First, the LPFC could bias the competition and reduce the intrusion of unwanted memory in later memory retrieval (*Kuhl et al., 2012*). Intentional suppression of memory retrieval reduces hippocampal activity via control mechanisms mediated by the LPFC (*Hulbert et al., 2016*). These studies suggest an important role of LPFC in control of memory. Second, studies examining retrieval-induced forgetting (*Anderson et al., 1994*; *Norman et al., 2007*) have found that retrieval practice could suppress (*Wimber et al., 2015*) or differentiate (*Hulbert and Norman, 2015*) the neural representation of competitive memories in the sensory cortices or hippocampus, and that these processes are mediated by the LPFC. Third, reactivation during wakefulness has been found to destabilize memories and has been associated with LPFC activation (*Diekelmann et al., 2011*), supporting its role in modifying reactivated memory representations.

The present study compared the neural reactivation during retrieval practice with that during restudy, and examined how the reactivated memory representations interact with the lateral PFC to achieve subsequent memory updating. In the Restudy condition, we simply asked subjects to study new replacement memories repeatedly. In the retrieval practice (RetPrac) condition, we asked subjects to retrieve the new replacement memory repeatedly. Feedback was provided as previous studies have shown that it could boost the behavioral performance (*Butler and Roediger, 2008*; *Pashler et al., 2005*). We hypothesized that during updating, retrieval practice would elicit greater reactivation of the outdated competitor (i.e., B) than would restudy exposures, reflecting the greater tendency of retrieval to trigger competition that requires correction. Despite these added difficulties, we predicted that during the final memory test, retrieval practice would lead to a better long-term effect: stronger reactivation of the replacement memory (i.e., the target, C) as well as weaker reactivation of the outdated competitor.

## Results

We used a multi-day design to examine both the short-term and long-term effects of memory updating under these two conditions. On Day 1, we extensively trained 19 subjects on associations between words (A) and pictures (B). On Day 2, we introduced new replacement A-C associations (B and C were of different visual categories) and asked subjects to replace the old associations with the new ones before entering the scanner. In the scanner, subjects encountered the new A-C associations three times, either via retrieval practice (RetPrac) or extra study exposures (Restudy). For

RetPrac, the cue word was paired with a black rectangle, and subjects were asked to recall the picture associated with the word. When the black rectangle turned red after 2 s, they were asked to judge the category of picture C by pressing one of the four buttons corresponding to Face, Object, Scene, and Don't know. Then the correct picture C was shown on the screen for 1 s as a feedback (*Figure 1A*, upper panel). For Restudy, the procedure was identical except the cue word was paired with the correct picture C at the beginning of the trial and subjects were asked to memorize the association and then make the category judgment. After these updating trials on Day 2, we waited for 24 hr to probe how successful retrieval practice and restudy were at accomplishing long-lasting memory updating. On Day 3, we scanned subjects again using a cued recall test. Subjects were asked to recall the visual details of the picture associated with the cue word presented on the screen, and then to respond by pressing one of the four buttons corresponding to Face, Object, Scene, and Don't know. They were then asked to perform a perceptual orientation judgment task for 8 s (*Figure 1A*, lower panel) before the next trial started.

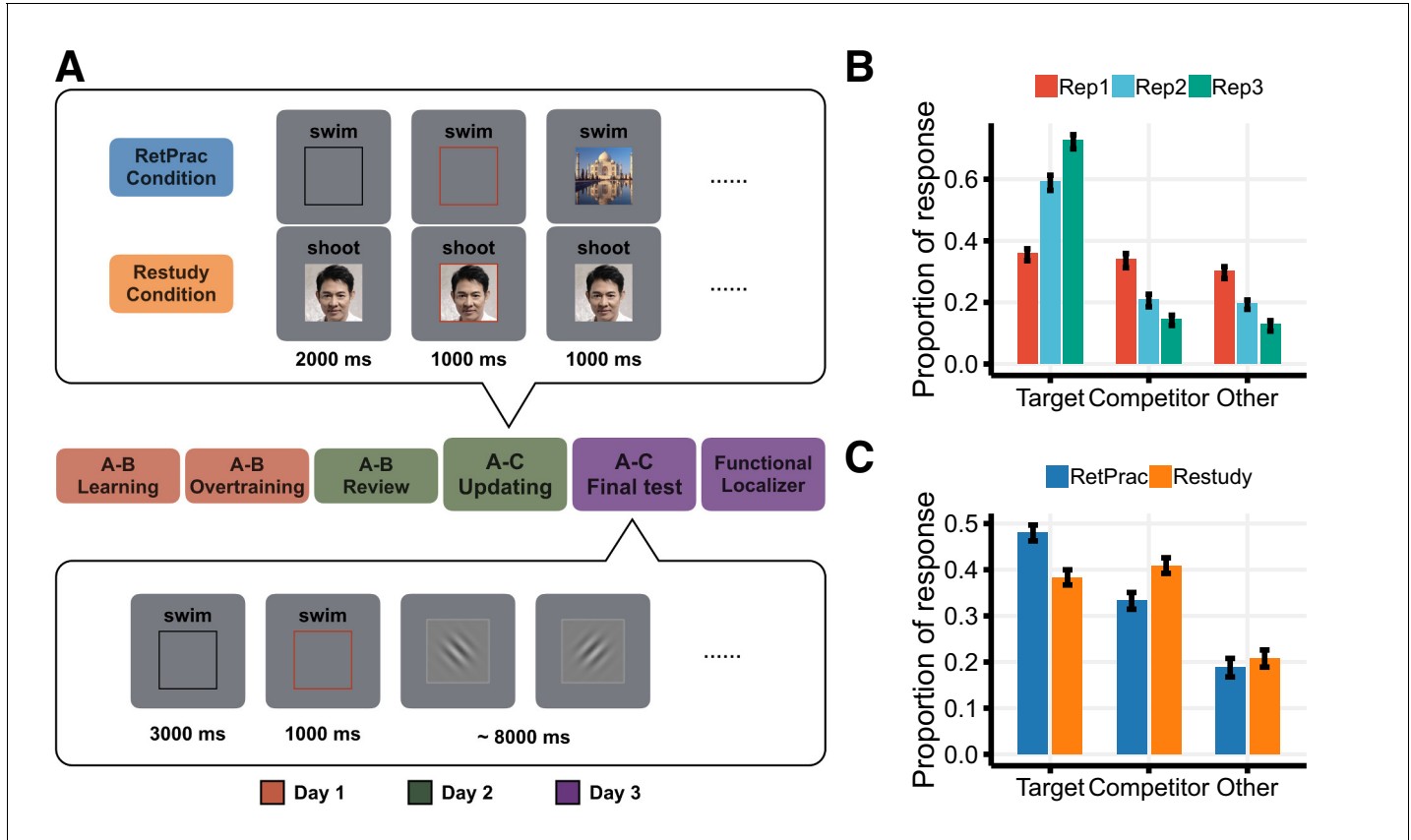

**Figure 1.** Experimental design and behavioral results. (**A**) Experimental design. On Day 1, subjects were over-trained with 144 pairs of word (A) – picture (B) associations. On Day 2, subjects were introduced to 144 new A-C associations (B and C were always from different visual categories) and asked to replace the old associations with the new ones before entering the scanner. In the scanner, each new A-C association was studied three times under one of the two updating conditions: Retrieval Practice (RetPrac) vs Restudy. Each trial started with a recall phase showing the cue word A paired either with a black rectangle (RetPrac) or with the associated picture C (Restudy). Two seconds after the recall phase, a red rectangle lasting for 1 s was shown and subjects needed to judge the category of picture C within this response window. Then the correct picture C was shown on the screen for 1 s as a feedback. On Day 3, subjects performed the A-C memory test while being scanned. The recall phase lasted for 3 s followed by a 1 s response window. Then subjects were asked to perform a perceptual orientation judgment task for 8 s. (**B**) Proportions of responses of targets (correctly choosing A-C), competitors (wrongly choosing A-B), and 'others' as shown according to each of the three study repetitions during updating. (**C**) The proportions of recalled targets, competitors, and 'other' categories during the final memory test one day after updating practice.

The online version of this article includes the following source data for figure 1:

**Source data 1.** Memory performance during Updating (Exp1).
**Source data 2.** Memory performance during final test (Exp1).

## Retrieval practice benefited long-lasting memory updating

Consistent with our hypothesis, we found that retrieval practice was a superior method for long-lasting memory updating. During the final memory test administered on Day 3, subjects recalled more updated targets ($t_{(18)}$ = 5.37, p<0.001, Cohen's d = 1.30, *Figure 1B*) with shorter response time ($t_{(18)}$ = −4.13, p<0.001, Cohen's d = 0.35), and showed fewer memory intrusions (i.e., accidental recall of the outdated competitors) ($t_{(18)}$ = −4.47, p<0.001, Cohen's d = 1.04) for pairs updated under the RetPrac condition compared to those updated under the Restudy condition. These findings indicate a significant self-testing effect in memory updating. Notably, the added difficulty of retrieval practice in the short term may have benefited the long-term test. During the acquisition of the replacement memory on Day 2, updating, retrieval practice performance was initially near the chance level, but it improved dramatically across the three practice repetitions, as indicated by increased ability to recall the target response category ($F_{(2, 36)}$=137.11, p<0.001), and by decreased production of responses in the competitor category ($F_{(2, 36)}$=9.82, p<0.001) or 'other' responses ($F_{(2, 36)}$=42.69, p<0.001) (*Figure 1C*). By contrast, behavioral performance for the restudy trials was near perfect across the three repetitions, reflecting the ease in processing material that is simply presented for re-study when no retrieval demand is involved (accuracy = 99.0%, and no difference across repetitions $F_{(2, 36)}$=0.42, p=0.66).

## Retrieval practice facilitated memory updating via differentiation

The above analysis suggests that retrieval practice could enhance the accessibility of A-C memories and reduce that of A-B memories in the A-C memory test. Several mechanisms could account for this result: (1) strengthening of A-C memory (*Karpicke and Roediger, 2008*; *Roediger and Butler, 2011*); (2) weakening of A-B memory (*Anderson et al., 1994*; *Levy and Anderson, 2002*); (3) differentiation of A-C and A-B memories (*Hulbert and Norman, 2015*; *Storm et al., 2008*). Previous studies suggest that reactivating the old memory and detecting and remembering the difference would help to resolve the proactive interference (*van den Honert et al., 2016*; *Wahlheim and Jacoby, 2013*). Thus, our results could be contributed by (4) integration and differentiation, which is a more specific version of the differentiation mechanism. To differentiate these hypotheses, we performed two additional behavioral experiments (Exp. 2 and 3) to examine how retrieval-practice affected A-B memory performance.

The inhibition hypothesis would predict greater C memory and weaker B memory intrusion in the A-C memory test, but worse B memory and greater C intrusion in the A-B memory test under the RetPrac condition than under the Restudy condition. By contrast, both the differentiation hypothesis and the integration and differentiation hypothesis would predict weaker B memory intrusion in the A-C memory test, and weaker or comparable C memory intrusion (given C memory was strengthen) in the A-B memory test.

In the new experiment (n = 46, Exp. 2), the procedure was nearly identical to the main experiment except that during the Day 3 test, we asked subjects to do both A-B and A-C memory tests, without the perceptual orientation judgment task between trials. To examine the effect of test order, half of the subjects did the A-B memory test first and the other half did the A-C memory test first. For both A-C and A-B memory tests and each response type (target, competitor, 'other'), test order (A-C first, A-B first) by update method (RetPrac, Restudy) two-way ANOVA revealed no significant main effect of test order (p-values>0.43, except for a trend towards a significant effect in 'other' responses during the A-B test, p=0.07, FDR corrected), nor of the test order by update method interaction (p-values >0.54, FDR corrected) (*Supplementary file 1*). Given our focus on the effect of updating method and the lack of updating method by test order interaction, we thus combined the data from both test order groups in the following analyses to increase the statistical power. Replicating the main effect, we found greater A-C memory under the RetPrac condition than under the Restudy condition during the A-C memory test, as indicated by the significantly higher correct recall of targets ($t_{(45)}$ = 3.93, p<0.001, Cohen's d = 0.57) and fewer competitor intrusions ($t_{(45)}$ = −2.75, p=0.008, Cohen's d = 0.36) under the RetPrac than under the Restudy condition (*Figure 2A*). However, we found that A-B memory was very strong during A-B recall (55.9% correct) and showed no effect of updating method on either B memory ($t_{(45)}$ = 1.43, p=0.19) or C intrusion ($t_{(45)}$ = 0.80, p=0.43).

This pattern was consistent with the differentiation hypothesis. That is, although C memory was strengthened by retrieval practice, no greater C intrusion was found in the A-B memory test. On the

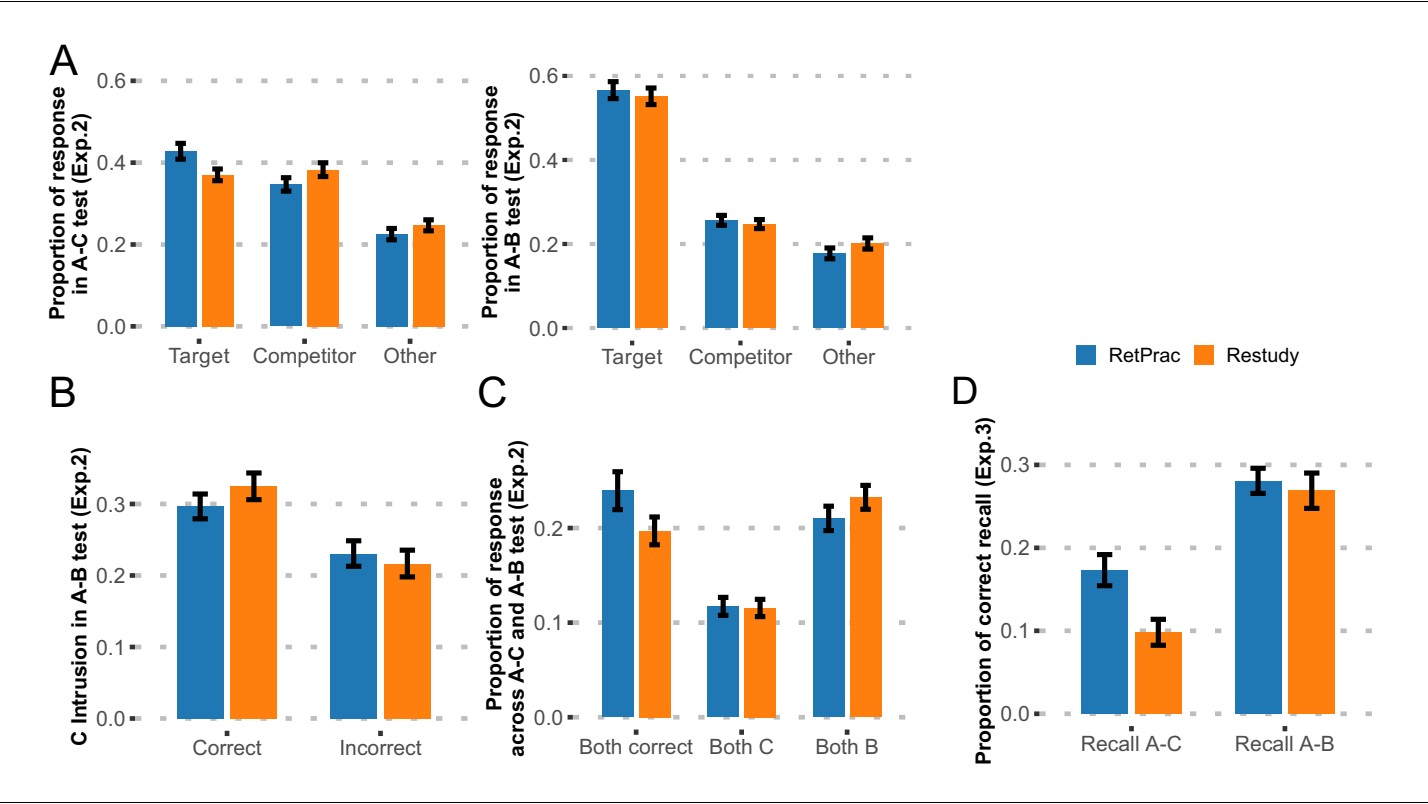

**Figure 2.** Results of follow-up behavioral Experiments 2 and 3. (A) Behavioral results from Exp. 2. The proportions of recalled targets, competitors, and 'other' categories during the final A-C and A-B memory tests one day after updating practice. Note, the targets and competitors were referred to as A-C and A-B memory in the A-C memory test, but as A-B and A-C memory in A-B memory test. (B) The proportions of A-C memory intrusions during A-B recall, as determined by whether the correct A-C memory was recalled during the A-C test. (C) Joint analysis for both A-B and A-C memories revealed memory differentiation. (D) Behavioral results from Exp. 3. The proportions of correct item recall during the final A-C and A-B memory tests one day after updating practice.

The online version of this article includes the following source data and figure supplement(s) for figure 2:

**Source data 1.** Memory performance during final test (Exp2).
**Source data 2.** Memory performance during final test (Exp3).
**Figure supplement 1.** Incorrect responses during updating and memory performance during the final test.

other hand, although B memory was comparable between the two conditions, there were fewer B intrusions in the A-C memory test. We noticed that the number of C memory intrusions was overall low, perhaps because of the weak A-C memory. We therefore did a further analysis to focus on strong A-C memory trials (correct trials in the A-C memory test), which would produce more intrusions. Consistently, we found that the correct trials in the A-C memory test (reflecting strong A-C memory) showed more C memory intrusions overall during the A-B test than did the incorrect trials ($t_{(45)}$ = 3.38, p=0.001, Cohen's d = 0.71). Interestingly, the intrusion rate was numerically smaller in the RetPrac condition than in the Restudy condition ($t_{(45)}$ = −1.59, p=0.12), which is consistent with the differentiation hypothesis (*Figure 2B*).

To further test the differentiation hypothesis, we did a joint analysis to examine the subjects' answers in both A-B and A-C memory tests given the same word cues. The differentiation hypothesis would predict more correct trials in both tests (i.e., C response in the A-C memory test and B response in the A-B memory test) under the RetPrac condition, i.e., that subjects maintained stronger and nonoverlapping representations of both A-B and A-C memories. Meanwhile, we would predict fewer trials in which subjects responded with old B memory in both tests (due to differentiation), and also fewer (due to differentiation) or comparable (due to strengthening of C and differentiation) trials in which subjects responded with new C memory in both memory tests. Our data supported all three predictions. We found that subjects made more correct responses in both

tests under the RetPrac condition than under the Restudy condition (24.0% vs 19.7%, $t_{(45)}$ = 3.81, p<0.001, Cohen's d = 0.48), but showed less B memory (21.0% vs 23.2%, $t_{(45)}$ = −2.61, p=0.012, Cohen's d = 0.25), and comparable C memory (11.7% vs 11.5%, $t_{(45)}$ = 0.18, p=0.86) in both memory tests (*Figure 2C*).

To further test whether RetPrac was able to modify the details of memory representation, in a third experiment (n = 28, Exp.3), we asked the subjects to write down the name of the associated picture (or any associated details if they could not recall the exact name) for each cue word, instead of choosing one of the three categories by pressing a button. Only answers with correct picture name or specific details were considered as a correct item recall. Our findings again replicated the retrieval-practice effect on A-C memory, as indicated by the significantly higher correct item recall ($t_{(27)}$ = 8.06, p<0.001, Cohen's d = 0.50) under the RetPrac condition than under the Restudy condition, but comparable performance on the A-B memory test ($t_{(27)}$ = 0.81, p=0.43) (*Figure 2D*). These results are consistent with the hypothesis that RetPrac helped to achieve better memory updating by differentiation and do not favor the idea that updating is accomplished by inhibition.

### The lack of suppression was not due to the weak proactive interference

Could the lack of suppression be due to the weak A-B intrusion? Subjects were well trained in the A-B association. However, we observed only slightly more competitor intrusions than unrelated errors (Exp. 1: 0.232 vs 0.209, $t_{(18)}$ = 1.93, p=0.069; Exp. 2: 0.233 vs 0.207, $t_{(45)}$ = 2.72, p=0.009; Exp. 3: 0.221 vs 0.200, $t_{(27)}$ = 2.33, p=0.027). This could reflect the fact that subjects were told explicitly that the associations had been changed and new associations should be explored and learnt, and is consistent with the differentiation account.

In additional analyses, we examined how the number of Day 2 intrusions was related to Day 3 performance. We found that pairs with more competitor intrusions during memory updating had worse A-C memory, more A-B intrusions, and comparable 'other' responses during the final A-C memory test. This pattern was consistent in both Exp. 1 and Exp. 2 (*Figure 2—figure supplement 1A,B*, left panel). Similarly, pairs with more 'other' responses during memory updating had worse A-C memory, more 'other' responses, and comparable A-B intrusions during the final A-C memory test (*Figure 2—figure supplement 1A,B*, right panel). These results suggested that the strength of proactive interference affected the new A-C learning. Importantly, there were many more A-B intrusions than 'other' responses during the final test, even when the number of responses during updating was matched, suggesting strong A-B interference (*Figure 2—figure supplement 1A,B*). Finally, we found that the number of competitor and 'other' responses during updating had no effect on A-B memory during the A-B memory test (*Figure 2—figure supplement 1C*), suggesting that the consolidated A-B memory was difficult to weaken. All of these results suggested that the lack of A-B memory suppression resulted from its strong representation.

### The effect of retrieval practice on target and competitor representations

To understand the neural basis of the retrieval-practice advantage, we used fMRI and MVPA to track the reactivation of the outdated and replacement memories during the two types of updating, and to link those patterns to memory performance on the final test. First, we trained neural classifiers to differentiate three categories of materials, i.e., faces, scenes, and objects, on the basis of independent functional localizer data. We then used them to examine the degree of memory reactivation during Day 2 updating and Day 3 final test (*Kuhl et al., 2012*; *Figure 3—figure supplement 1A*). We focused our analysis on the medial prefrontal cortex (MPFC), ventral temporal cortex (VTC), angular gyrus (AG), and hippocampus (HPC), which overlap with the core recollection network (*Rugg and Vilberg, 2013*) and have consistently shown a neural reinstatement effect during memory retrieval (*Kuhl and Chun, 2014*; *Wimber et al., 2015*; *Xiao et al., 2017*). Confirming the relevance of these regions to our task, we found that the classifier output could predict subjects' categorical judgments during the final test with significantly above-chance accuracy in the MPFC, VTC, and AG (ranging from 39.6% to 42.6%, all p-values <0.001, all survived FDR correction), but not in the HPC (35.5%, p=0.35) (*Figure 3—figure supplement 1B*). The following analysis thus focused on the first three regions (*Figure 3A*).

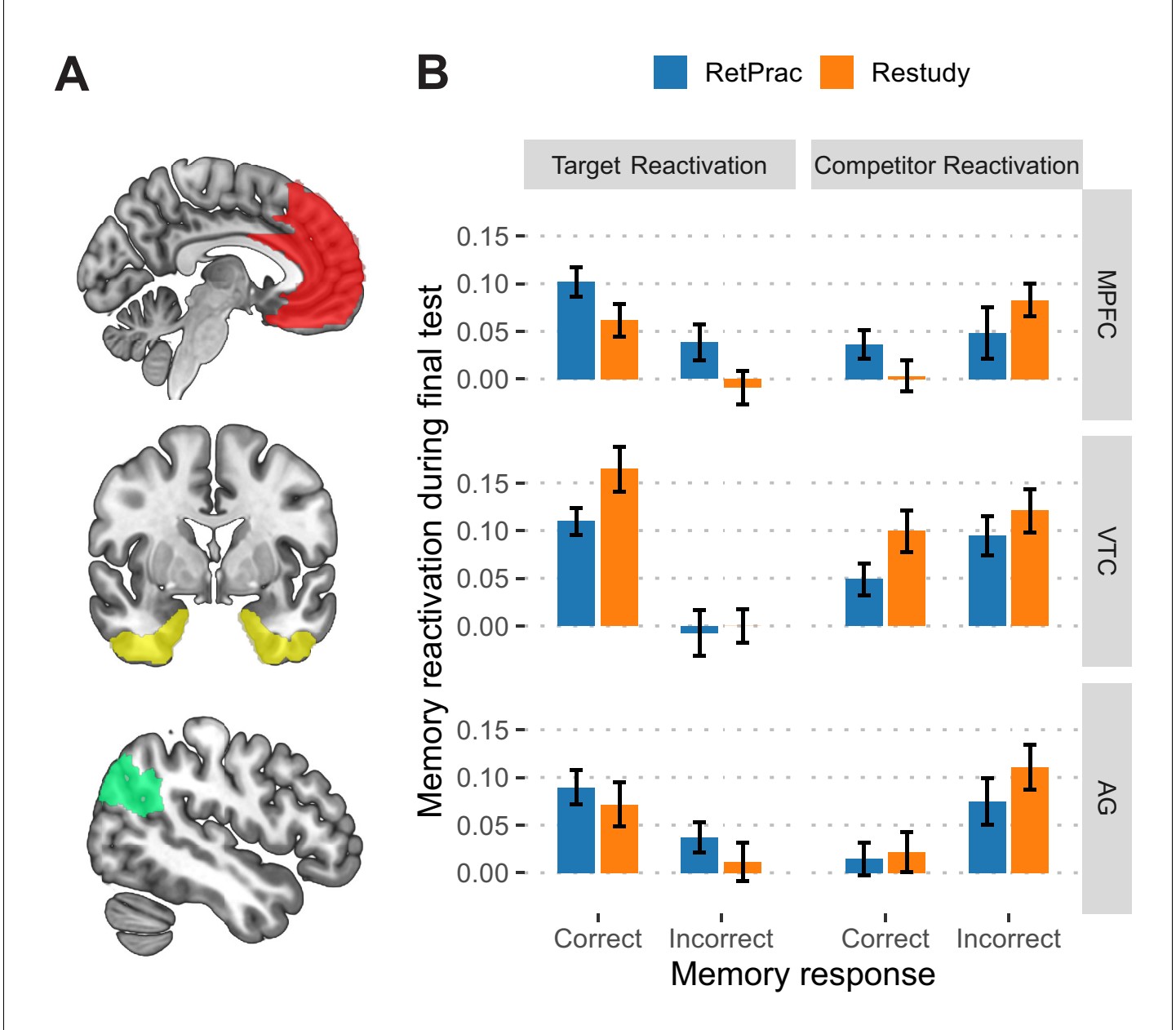

**Figure 3.** Neural reactivation during the final memory test. (**A**) Depiction of the anatomical ROIs used in the main analysis. All ROIs consisted of regions from both hemispheres. (**B**) The reactivation of target (picture C) and competitor (picture B) during the final test as a function of updating method and memory outcome, based on classifier outputs (after subtracting 'other' evidence). Error bars indicate within-subject standard errors.

The online version of this article includes the following source data and figure supplement(s) for figure 3:

**Source data 1.** Classifier evidence during the final test.
**Figure supplement 1.** Classifier performances.

Next, we used these classifiers to examine how retrieval practice could shape the target and competitor representations in the brain during the final memory test. To further examine how the representations in these regions were differentially modulated by retrieval practice and behavioral performance, we separately examined the correct trials (i.e., when targets were chosen) and the incorrect trials (i.e., when competitors were chosen). Three-way repeated-measures ANOVA was conducted, with evidence (Target vs Competitor), outcome (Correct vs Incorrect) and updating method (RetPrac vs Restudy) as within-subject factors. Both the inhibition and differentiation

hypothesis would predict stronger target representation and weaker competitor representation for RetPrac than for Restudy, whereas the integration and differentiation hypothesis would predict strong reactivation of both target and competitor representations for RetPrac, despite the superior behavioral performance.

In the MPFC, three-way ANOVA revealed a significant evidence-by-outcome interaction ($F_{(1,18)}$ = 23.25, p=0.0001, survived FDR correction, *Figure 3B*), suggesting that the activation in the MPFC tracked behavioral performance. We also found a significant method-by-evidence interaction ($F_{(1,18)}$ = 6.50, p=0.02, survived FDR correction). No three-way interaction or method-by-outcome interaction was found (p-values >0.08, *Supplementary file 2a*). We then did two separate two-way ANOVAs for target and competitor evidence reactivation. For target reactivation, there were significant main effects of updating method (RetPrac vs Restudy) ($F_{(1,18)}$ = 7.01, p=0.02, *Supplementary file 2b*) and outcome (chosen targets vs chosen competitors) ($F_{(1,18)}$ = 15.10, p=0.001, survived FDR correction), suggesting that correct responses were associated with stronger target evidence reactivation, and that retrieval practice was able to boost the target reactivation for both correct and incorrect trials. For competitor evidence, however, we found that there was stronger competitor evidence reactivation for incorrect trials than for correct trials under the Restudy condition ($t_{(18)}$ = 3.46, p=0.003, Cohen's d = 0.86, survived FDR correction), whereas no such difference was found for the RetPrac condition ($t_{(18)}$ = 0.35, p=0.73), although the outcome-by-condition interaction did not reach significance ($F_{(1,18)}$ = 2.53, p=0.13, *Supplementary file 2b*). The latter result indicated that even when correct responses were made, there was still strong and comparable competitor reactivation under the RetPrac condition, suggesting that RetPrac integrated and differentiated competitor and target evidence in the MPFC.

In the AG, three-way ANOVA also revealed significant a evidence-by-outcome interaction ($F_{(1,18)}$ = 18.46, p=0.0004, survived FDR correction, *Figure 3B*), suggesting that the representation in the AG tracked behavioral performance. We also found a significant method-by-evidence interaction ($F_{(1,18)}$ = 7.45, p=0.01, survived FDR correction). No other main effect or interaction was found (p-values >0.36, *Supplementary file 2a*). Once again, we performed two separate two-way ANOVAs for target and competitor reactivation. For target reactivation, there was a significant main effect of response type ($F_{(1,18)}$ = 7.23, p=0.02, survived FDR correction, *Supplementary file 2b*), with stronger target evidence for correct responses than for incorrect responses. For competitor reactivation, there was a significant main effect of response type ($F_{(1,18)}$ = 6.04, p=0.02, *Supplementary file 2b*), with stronger competitor evidence for incorrect responses than for correct responses. Together, these results suggested that the AG representation mainly tracked the behavioral performance.

In the VTC, we also found a significant evidence-by-outcome interaction ($F_{(1,18)}$ = 26.51, p<0.0001, survived FDR correction, *Figure 3B*), again suggesting that the representation in the VTC tracked behavioral performance. Interestingly, we found a significant main effect of method ($F_{(1,18)}$ = 5.58, p=0.03), which did not interact with other factors (p-values >0.20, *Supplementary file 2a*), suggesting that retrieval practice significantly suppressed competitor evidence ($F_{(1,18)}$ = 4.68, p=0.04) and marginally reduced the target evidence ($F_{(1,18)}$ = 3.26, p=0.09) in the VTC.

## Retrieval practice and restudy were associated with distinct subsequent memory effects

The improved memory that arises from retrieval practice may be supported by neural mechanisms that are distinct from those involved in restudy, and the former mechanisms could produce more resilient traces than the latter. To test this possibility, we performed an analysis to determine whether the two updating methods were associated with different subsequent memory effects. In particular, we examined the pattern of target reactivation during updating and subsequent memory.

This analysis revealed distinct patterns for the Restudy and RetPrac conditions: correctly recalled items (i.e., target) showed stronger target activation than did incorrectly recalled ones (i.e., competitor) in the VTC ($t_{(18)}$ = 4.29, p<0.001, Cohen's d = 0.79, survived FDR correction) during restudy (all other ROIs, p-values >0.13), whereas retrieval practiced items that were later correctly recalled were associated with stronger target activation than incorrectly recalled ones in the MPFC ($t_{(18)}$ = 2.66, p=0.016, Cohen's d = 0.60, survived FDR correction, *Figure 4A*) (all other ROIs, p-values >0.12). This finding suggests that successful memory updating may involve different representations under the RetPrac and Restudy conditions. The greater association of MPFC representation with enduring retention is consistent with its putative involvement in the consolidation process (*Antony et al.,*

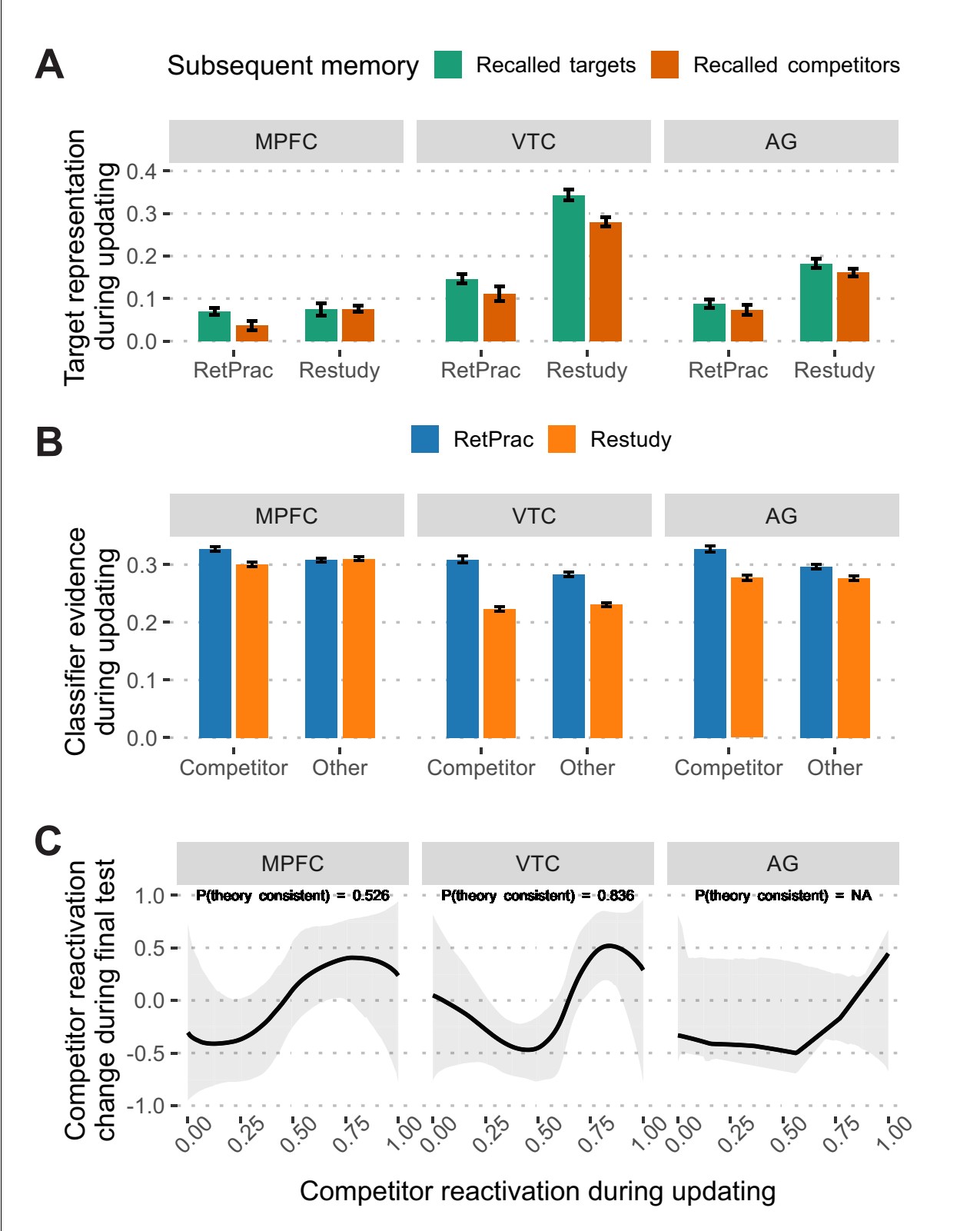

**Figure 4.** Memory reactivation during updating and its change in the final test. (**A**) Target memory representation (after subtracting 'other' evidence) during updating as a function of subsequent memory performance (correctly recalled targets vs incorrectly recalled competitors) during the final memory test. (**B**) Classifier evidence of competitor and other categories during the A-C updating phase under the Restudy and RetPrac conditions. Restudy was associated with weaker competitor reactivation. (**C**) Model fitting of the nonmonotonic plasticity hypothesis under the RetPrac condition.

*Figure 4 continued on next page*

*Figure 4 continued*

Only VTC showed the hypothesized pattern in which modest competitor reactivation (normalized into [0, 1] range) weakened, and strong competitor reactivation enhanced later competitor memory reactivation. Error bars indicate within-subject standard errors.

The online version of this article includes the following source data and figure supplement(s) for figure 4:

**Source data 1.** Classifier evidence during memory updating by subsequent memory performance.
**Source data 2.** Classifier evidence during memory updating.
**Figure supplement 1.** Classifier evidence of competitor and 'other' categories for correct trials during the A-C updating phase.
**Figure supplement 2.** Change of competitor and target evidence with training under the RetPrac condition.
**Figure supplement 3.** Nonmonotonic plasticity model fitting the results for the Restudy condition.

*2017*; *Tompary and Davachi, 2017*), and suggests that retrieval practice improves updating by driving consolidation more successfully than does restudy.

## Retrieval practice was associated with greater competitor reactivation during updating

In addition to the greater engagement of MPFC, retrieval practice advantage on Day 3 might also derive in part from the need to overcome retrieval competition during the updating on Day 2. To test this hypothesis, we looked at mnemonic representations for competitors during the updating process, although the competitors were not presented under either the RetPrac or the Restudy conditions. As predicted, we found significant competitor reactivation (compared with 'other' evidence) under the RetPrac condition in the MPFC ($t_{(18)}$ = 3.26, p=0.004, Cohen's d = 0.98), VTC ($t_{(18)}$ = 3.20, p=0.005, Cohen's d = 1.03), and AG ($t_{(18)}$ = 4.31, p<0.001, Cohen's d = 1.34), all survived FDR correction (*Figure 4B*). By contrast, there was no evidence of reactivation of the outdated competitors in any of the three ROIs in the Restudy condition (all p-values >0.13). Direct comparisons revealed significantly stronger competitor reactivation under the RetPrac condition than under the Restudy condition in the MPFC ($t_{(18)}$ = 4.32, p<0.001, Cohen's d = 1.36), VTC ($t_{(18)}$ = 10.87, p<0.001, Cohen's d = 3.46), and AG ($t_{(18)}$ = 6.98, p<0.001, Cohen's d = 2.21), again all survived FDR correction. A similar pattern was found when only correct (target) trials were included (all p-values <0.017 for the RetPrac condition; all p-values >0.07 for the Restudy condition; all p-values <0.005 when directly comparing the RetPrac and Restudy conditions; *Figure 4—figure supplement 1*).

The targets were only briefly presented as feedback under the RetPrac condition, whereas they were shown throughout the whole trial under the Restudy condition, so the differences in trial structure might bias the classifier performance. In particular, the reduction in competitor evidence under the Restudy condition might be due simply to the strong target evidence accumulated over a longer interval, but not to the lack of competitor reactivation. This possibility predicts lower evidence not only for the competitor, but also for the 'other' (third) category under the Restudy condition. Contrary to this baseline shift hypothesis, we found no significant differences in the evidence for the 'other' category between the RetPrac and the Restudy conditions in MFPC ($t_{(18)}$ = −0.42, p=0.68). We did, however, find evidence for a baseline shift in the VTC ($t_{(18)}$ = 9.25, p<0.001) and AG ($t_{(18)}$ = 3.60, p=0.002) (*Figure 4B*), suggesting that reduced competitor activation in that structure during restudy may be due in part to differences in trial structure. There was, however, significant evidence (Competitor vs Other) by updating method (RetPrac vs Restudy) interaction in all three regions (all p-values <0.01), indicating that retrieval practice, as a method of updating, elicited significantly greater competition from distracting representations. This additional competition posed extra difficulties that needed to be overcome, difficulties that did not arise during restudy. This finding corresponds well with behavioral evidence of the increased incidence of competitor intrusions during retrieval practice, relative to during restudy.

Across the three repetitions, we found that the target evidence increased with the number of repetitions in the VTC ($t_{(18)}$ = 3.44, p=0.003), but not in the MPFC ($t_{(18)}$ = 1.75, p=0.10) or AG ($t_{(18)}$ = 1.78, p=0.09) (*Figure 4—figure supplement 2*). Strikingly, although subjects made fewer competitor responses across repetitions, we did not find a significant reduction in competitor evidence across repetitions (all p-values >0.73). This result fits very well with the integration and differentiation hypothesis.

## Reactivation-dependent memory updating in VTC during retrieval practice

The analyses so far revealed that compared to restudy, retrieval practice led to greater competitor reactivation during updating, but to reduced competitor reactivation during the final test in the VTC, but not in the MPFC. These findings are consistent with the possibility that retrieval practice may drive memory suppression in the VTC but not MPFC, triggered by their reactivation during the updating process. According to the nonmonotonic plasticity hypothesis, there is a nonlinear relationship between the strength of the memory reactivation and its later change, such that moderate reactivation has a weakening effect whereas strong reactivation has a strengthening effect (*Kim et al., 2014*; *Newman and Norman, 2010*; *Ritvo et al., 2019*). If this is the case, we would predict a U-shaped relationship between competitor reactivation strength during updating and during the final test.

To test this hypothesis, we used the P-CIT Bayesian curve-fitting algorithm to estimate the shape of the curve between competitor evidence during updating and the final test (*Detre et al., 2013*). The model hypothesizes a U-shaped curve, and the posterior probability of the theory consistency [P(Theory consistent)] indicates how well the fitted curve aligns with this hypothesis (the greater, the better, chance level = 0.5). We fitted this model in all three ROIs separately. The results suggest that VTC showed a pattern that was highly consistent with the model under the RetPrac condition [P(Theory consistent)=0.836, p=0.012, *Figure 4C*]. No such pattern was found in the MPFC [P(Theory consistent)=0.526] and the model fitting failed for the AG. All model fitting in these three regions failed under the Restudy condition (*Figure 4—figure supplement 3*). This may be the result of weak competitor reactivation during restudy, which did not cover the full range of the nonmonotonic plasticity curve.

Together, the results suggested that competitor reactivation was associated with subsequent suppression, which could be attributed to nonmonotonic synaptic plasticity. However, we found this effect in only the VTC and not in the MPFC or AG, suggesting region-specific effect of memory suppression.

## The LPFC contributed to MPFC memory updating under the RetPrac condition

To identify processes that contributed to goal-directed modulation of reactivated memories, we compared brain activity during updating between the RetPrac and Restudy conditions. We found that updating by retrieval practice engaged the left lateral prefrontal cortex (LPFC), dorsal anterior cingulate gyrus (dACC), bilateral anterior insular cortex (AI), and caudate nucleus (*Figure 5A*, *Supplementary file 3a*) more than did updating by restudy, whereas the hippocampus and other regions showed significantly weaker activation (*Figure 5—figure supplement 1*, *Supplementary file 3b*). These findings are consistent with prior work showing the engagement of cognitive control during retrieval (*Wimber et al., 2015*). To further probe the function of these regions in overcoming intrusions from outdated competitors, we examined how these prefrontal and striatal activations varied with updating performance during retrieval practice. We distinguished between those retrieval-practiced trials that subjects recalled incorrectly (incorrect trials, IC), those they got correct the first time a given pair was shown (First Correct, FC), and those that they got correct the second or third time for a given pair was shown (Later Correct, LC). The rationale is that compared to LC trials. IC and FC trials should have a greater need for competition resolution. In addition, the FC trials may engage stronger reward-based learning than IC and LC trials because of the former's greater positive prediction error. Both mechanisms could contribute to representational change and memory differentiation.

Consistently, we found that the left LPFC activation for LC trials was significantly lower than that for FC trials ($t_{(18)}$ = −7.17, p<0.001) and IC trials ($t_{(18)}$ = −4.76, p<0.001) (*Figure 5B*), suggesting that the left LPFC activation was mainly driven by the extent of memory competition. A similar pattern was also found in the dACC and AI (*Figure 5—figure supplement 2*, *Supplementary file 3c*). By contrast, the caudate activation for the FC trials was significantly greater than that for the other two types of trials (all p-values <0.004, *Supplementary file 3c*), consistent with its role in prediction-error-based processing.

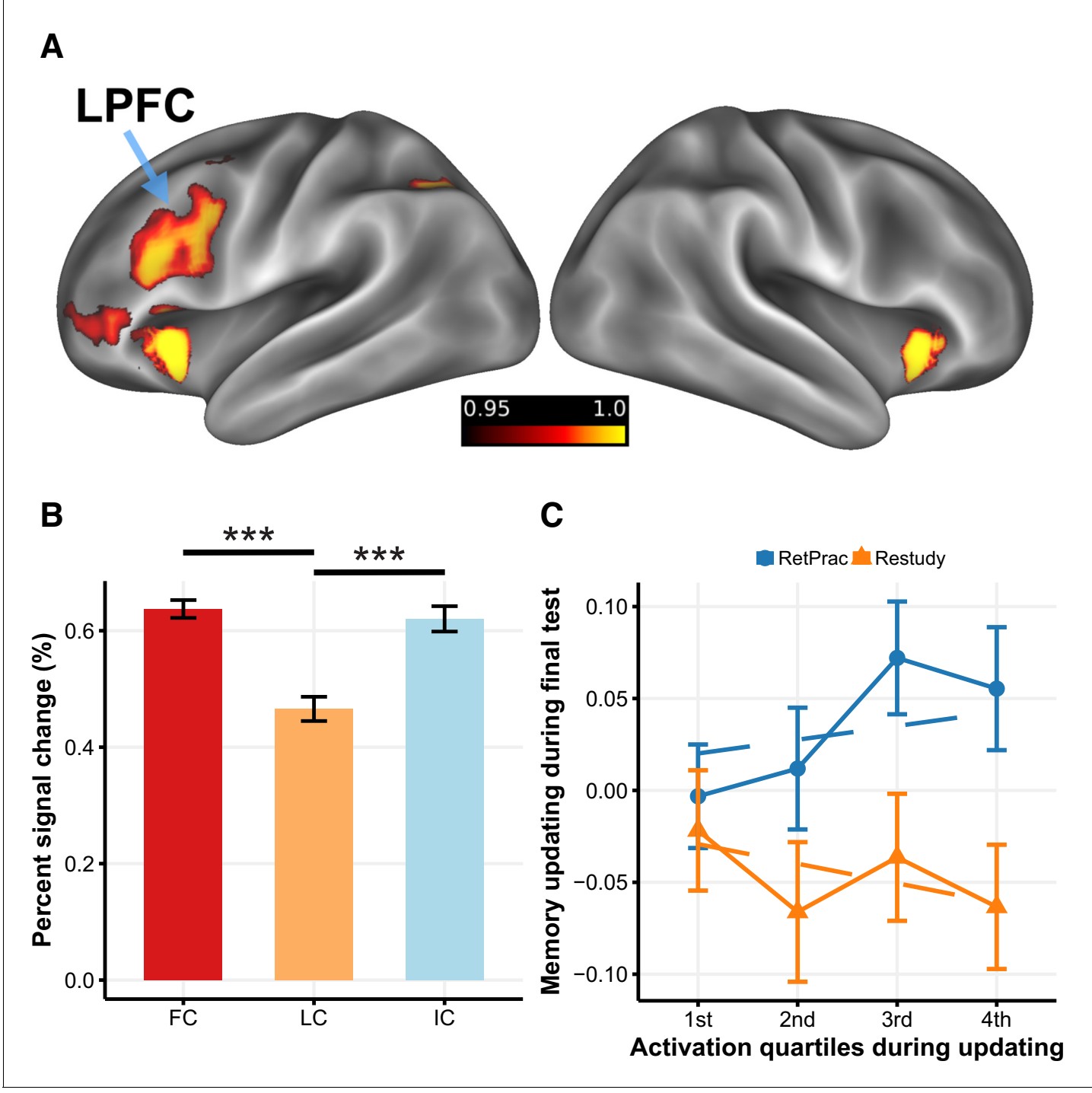

**Figure 5.** LPFC activity and memory updating under the RetPrac condition. (**A**) Brain regions that showed greater activation during memory updating under the RetPrac than under the Restudy condition. The color bar indicates one minus the P-value (corrected). (**B**) Activity in the LPFC was sensitive to pairs' updating performance. The failed recall (IC) and the successfully recalled the first time (first correct, FC) trials showed greater activation than the successfully recalled the second or the third time (later correct, LC) trials, indicating that the LPFC was involved in inhibiting competitive memories. (**C**) MPFC memory updating (target representation minus competitor representation) as a function of LPFC activation during updating (divided by quartiles). Error bars indicate within-subject standard errors.

The online version of this article includes the following source data and figure supplement(s) for figure 5:

**Source data 1.** Percent signal change by conditions during memory updating.

**Figure supplement 1.** Brain regions showing greater activation for restudy than retrieval practice.

**Figure supplement 2.** Brain activity as a function of memory performance.

To link the LPFC and caudate activation to representational change during updating, we examined whether the caudate and LPFC activation in the current repetition was related to competitor suppression in the subsequent repetition. Owing to the caudate's role in prediction error, we focused on the first correct trial (FC). This revealed that strong caudate activation was associated with greater competitor evidence reduction in the next repetition in the VTC ($\chi^2_{(1)}$=5.86, p=0.015), but not in MPFC or AG (all p-values >0.15), suggesting that the VTC evidence could be temporary weakened by reinforcement learning. No such effect was found for LPFC when focusing on the incorrect and first correct trials (p-values >0.41), suggesting that LPFC did not temporally suppress the competitor evidence.

We further examined whether the LPFC and caudate activity during retrieval practice was associated with long-term memory updating on Day 3. This analysis revealed that trials with greater LPFC activity during retrieval practice ultimately showed superior memory updating (i.e., target – competitor evidence) during the final test on Day 3 in the MPFC ($\chi^2_{(1)}$=4.62, p=0.032), but no effect was found in the caudate. Together, these results suggest that the LPFC is involved in resolving retrieval competition between targets and competitors, and ultimately contributes to successful long-term memory updating in the MPFC. By contrast, the caudate may suppress short-term representation through reward-based supervised learning.

## Discussion

Memory updating serves an adaptive role in ensuring that the most relevant information is accessible in memory. Behavioral studies have long emphasized the role of retrieval in memory updating, yet the neural mechanisms behind this process are barely understood. We found that, compared to simple restudy, retrieval practice was associated with better memory updating without suppressing the old memories. Furthermore, by tracking the neural evidence of old and new memories during both final memory and updating, we demonstrated that superior memory updating under retrieval practice could be achieved by multiple mechanisms. These results provide important insights into the neural mechanisms of memory updating.

When updating memory with replacement information, one needs to enhance the new target memory, inhibit the outdated competing traces, and/or differentiate the old and new memory traces. These different mechanisms could be examined by testing A-B memory. We found that retrieval practice had no effect on A-B memory, albeit it significantly enhanced new memory and reduced old memory intrusions in the A-C memory test. Further supporting the differentiation hypothesis, retrieval practice increased the number of trials in which subjects appropriately chose the targets in different test conditions (an indication of differentiation), and reduced the number of the trials in which the same responses were made in both test conditions (an indication of indifferentiation). At least two factors might contribute to the lack of suppression of old memory trace. First, some studies have shown that retrieval-induced forgetting was more pronounced after a short delay (minutes to hours) than after a long delay (days) (*Abel and Bäuml, 2014*; *Liu and Ranganath, 2019*; *Murayama et al., 2014*). Second, the old memory was extensively trained and consolidated, which made it harder to inhibit. In any case, our results suggest that reduced intrusions could be achieved without significantly suppressing the old memories, but by strengthening the new memory traces and differentiating the old and new memory traces.

The current study revealed several neural mechanisms that could account for the advantages of retrieval practice in memory updating. First, we found that retrieval practice could shift the neural substrates from VTC to MPFC, which is involved in fast system consolidation (*Antony et al., 2017*). Existing studies have shown that during retrieval, item-specific reactivation is generally not found in the VTC (*Favila et al., 2018*; *Xiao et al., 2017*). As a result, with repeated retrieval practice, the brain may rely less on the sensory information for mnemonic decisions. These features are consistent with the behavioral findings that retrieval practice does not improve the quality of sensory memory (*Sutterer and Awh, 2016*), and may promote gist-based false memory (*McDermott, 2006*). Our study suggests that the MPFC may be responsible for this gist-based memory given its role in schema-based learning (*Gilboa and Marlatte, 2017*; *Preston and Eichenbaum, 2013*).

Second, feedback was provided during retrieval practice in the current study. Although existing studies have found a significant effect of retrieval practice when no feedback was provided (*Karpicke and Roediger, 2008*), feedback has been consistently shown to improve memory

performance (*Butler and Roediger, 2008*; *Pashler et al., 2005*). The current study also found greater caudate activation under the RetPrac condition, in particular for the first correct trial, which is consistent with its role in processing positive prediction error (*O'Doherty et al., 2004*). Recent studies suggest that prediction error plays an important role in memory updating (*Kim et al., 2014*) and reconsolidation (*Lee et al., 2017*), and that the caudate is involved in modifying and re-encoding the retrieved memory representation (*Scimeca and Badre, 2012*). Extending these observations, we found that the caudate's activation during first correct response was associated with reduced competitor evidence in the visual cortex, lending support to the idea that the supervised-learning mechanism could lead to representational changes (*Ritvo et al., 2019*).

Third, retrieval practice is an effortful process as compared to simple restudy. Consistently, neuroimaging studies have found greater neural activity during retrieval than during restudy (*Wing et al., 2013*). In addition, retrieval practice could also potentiate subsequent learning, which is associated with greater frontoparietal activity (*Nelson et al., 2013*). Our results are highly consistent with these observations, revealing stronger activation in the LPFC, dACC and insula. We further found that LPFC activation was greater when there was a greater competitor intrusion, which is consistent with its role in controlled memory retrieval among competitors (*Badre et al., 2005*). Previous studies have further implicated the LPFC in reducing the intrusion of competitors in memory retrieval (*Kuhl et al., 2012*), and in reducing competition memories through cortical pattern suppression (*Wimber et al., 2015*). The current study did not find a strong association between LPFC activation and competitor suppression during updating, possibly because the to-be-suppressed competing memories in the current study were well trained and consolidated by overnight sleep. We however found that the LPFC activation was associated with long-term memory updating in the MPFC, suggesting that the LPFC might help to resolve the interference between old and new memories.

Critically, the current study identified a region-specific relationship between competitor reactivation during updating and later memory changes. Consistent with previous studies (*Kuhl et al., 2012*; *Wimber et al., 2015*), we found significant VTC competitor reactivation during updating and competitor suppression during the final test. One difference is that the current study also found a trend of target suppression in this region, whereas previous studies found target enhancement during retrieval practice. Furthermore, we found a U-shaped relationship between competitor reactivation in the VTC during updating and the final test, which is consistent with the nonmonotonic plasticity mechanism (*Ritvo et al., 2019*).

A different pattern was found in the MPFC, where the target evidence was strengthened. More importantly, it was integrated but differentiated from the competitor evidence, as indicated by the comparable competitor reactivation for correct and incorrect responses. Furthermore, we found that the MPFC's memory updating was not predicted by the nonmonotonic plasticity principle, but was rather associated with LPFC activity driven by the competitor reactivation. The MPFC has been implicated in memory integration and updating (*Preston and Eichenbaum, 2013*; *Zeithamova et al., 2012*). Our results replicated and extended these observations by showing that memory integration could occur even when competing memories were simultaneously reactivated. This fits very well with the hypothesis that MPFC is able to develop rapidly integrated neocortical representations of reactivated memory traces during retrieval practice (*Antony et al., 2017*).

These results suggest that during retrieval practice, the co-activation of old and new memories might provide a unique opportunity to modify these representations and to facilitate memory updating. The MPFC could form integrated representations of co-activated and competing memories, while the LPFC control mechanism might contribute to memory updating by selectively strengthening the reactivated target memory and differentiating the old and new memory representations in the MPFC. The differentiation could be achieved by adding contextual representations into the memory trace, thus forming more unique representations of old and new memories, and/or by linking old and new memory representations to different aspects of cue representations. These processes could be further enhanced by feedback-driven supervised learning during retrieval practice, as well as by non-supervised Hebbian learning that involves nonmonotonic plasticity (*Ritvo et al., 2019*). Future studies should further examine how the LPFC and MPFC could contribute to the representational change of the reactivated old and new memories.

In the current study, we also found significant memory reactivation in the angular gyrus during both updating and the final test. Unlike memory reactivation in the VTC or MPFC, we found that memory reactivation in the angular gyrus tracked closely the behavioral performance. Consistently,

other studies have shown that the angular gyrus exhibits abstract yet item-specific mnemonic representations (*Kuhl and Chun, 2014*; *Xiao et al., 2017*), which is modulated by mnemonic goals (*Favila et al., 2018*). Through its connection with the more anterior region, i.e., the lateral intraparietal sulcus (latIPS), the representations in the angular gyrus can serve as a mnemonic buffer that helps to make mnemonic decisions (*Sestieri et al., 2017*; *Wagner et al., 2005*). Our results support the idea that AG functions as a multimodal convergent zone to combine memory signals from multiple brain regions and to form a memory representation that is closely related to subjective experience and memory decisions.

Successful memory retrieval is often associated with greater hippocampal activity. Interestingly, we observed weaker hippocampal activation under the RetPrac condition than under the Restudy condition. Previous studies suggested that the hippocampus might be inhibited when subjects were required to suppress thoughts and memories (*Benoit and Anderson, 2012*; *Hulbert et al., 2016*). It is thus tempting to speculate that the hippocampus might be inhibited as a result of strong competition from the old memory. The deactivation itself, however, might not be sufficient to support the inhibition hypothesis. For example, although the MPFC also showed weaker activation under the RetPrac condition, MVPA analysis suggested that the MPFC represented task-related information that was related to subsequent memory performance (*Figure 4A*), suggesting that it played an important role in memory updating. Several major factors might account for the chance-level decoding of memory information in the hippocampus during retrieval. On the one hand, the classifier accuracy during training was lower in the hippocampus than in other regions. This might be due to the sparse nature of hippocampal representation (*Quiroga et al., 2008*), the low signal-to-noise ratio in this region, and/or the weak categorical representation. Consistently, previous studies also found hippocampal representation when using representational similarity analysis to probe item-level representations (e.g., *Jonker et al., 2018*; *Tompary and Davachi, 2017*; *Wimber et al., 2015*; *Xiao et al., 2017*). On the other hand, we trained the classifiers during perception and applied them to memory retrieval. Previous studies have shown that memory representation could be transformed from encoding to retrieval (*Chen et al., 2017*; *Xiao et al., 2017*; *Xue, 2018*), and that this transformation could have further reduced the classifier performance (*Albers et al., 2013*). Future studies should use an optimized design and item-level analysis to further elucidate the role of the hippocampus in memory updating.

Memory is a dynamic process that depends on memory reactivation. On the one hand, reactivation of a target memory will strengthen the activated memory (*Tambini and Davachi, 2019*; *Xue, 2018*). On the other hand, reactivation of unwanted memories (e.g., competitors) can facilitate the suppression of those unwanted memories (*Lee et al., 2017*). Beyond the mechanisms of strengthening and weakening memory representations to achieve memory updating, the current study adds to the growing literature reporting that reactivation (via retrieval practice) can also integrate and differentiate co-activated memories (*Chan et al., 2009*; *Hulbert and Norman, 2015*; *Schlichting et al., 2015*; *Zeithamova et al., 2012*) and hence can increase the flexibility of memory updating in different contexts. A better understanding of these diverse mechanisms can be leveraged to develop more effective behavioral interventions to modify maladaptive memories in some psychiatric conditions.

## Materials and methods

### Subjects

Nineteen healthy college students (seven males; mean age = 21.5 years, range = 18–25 years) participated in the fMRI study, and two additional samples of 46 (18 males; mean age = 22.6 years, range = 18–29 years) and 28 college students (five males; mean age = 21.0 years, range = 18–24 years) participated the two behavior experiments (Exps 2 and 3), respectively. The sample size of the fMRI study was comparable with that in several previous studies using similar paradigm (*Kuhl et al., 2012*; *Wimber et al., 2015*). All subjects had normal or corrected-to-normal vision, and no history of psychiatric or neurological diseases. Three additional subjects were recruited into the fMRI study but excluded from the final analysis due to either scanner malfunction or chance-level memory performance. Written consent was obtained from each subject after a full explanation of the study

procedure. The study was approved by the Institutional Review Boards at Beijing Normal University and the Center for MRI Research at Peking University.

## Materials

Stimuli consisted of 144 Chinese words (cues) and 288 pictures (associates). All words were two-character Chinese verbs. Pictures were color photographs from three categories, including famous faces (e.g., Jet Li), common objects (e.g., toothbrush), and famous scenes (e.g., the Forbidden City). Each word was associated with two pictures from different categories (e.g., A-B, word-face associations; A-C, word-object associations). Half of the words were assigned to the RetPrac condition and the other half to the Restudy condition. The assignment was counterbalanced between subjects.

## Experimental procedure

The whole experiment lasted for three consecutive days. On Day 1, subjects were trained on the 144 word-picture associations (A-B learning). On Day 2, subjects were asked to update the old memory with the new A-C associations, half under the RetPrac condition and the other half under the Restudy condition. On Day 3, subjects performed a cued recall task to test A-C associations.

### A-B learning

Before learning, subjects were asked to view all 288 pictures and their corresponding labels to make sure that each picture was correctly identified. Subjects were then instructed to learn 144 word-picture associations. Each A-B association was presented for 4 s after a 0.5 s fixation, and subjects were asked to memorize the associations for a later test. After the initial learning, subjects went through an overtraining phase. For each trial, the cue word and a black rectangle were presented for 2 s. Subjects were instructed to recall the details of the picture associated with the cue word presented on the screen. The rectangle turned to red for 1 s and subjects needed to indicate their responses by pressing the button corresponding to the picture's category (Face, Object, Scene, and Don't Know). The correct B picture was presented for 1 s as a feedback and subjects were instructed to use this feedback to further strengthen their memory. All A-B associations were tested and those correctly recalled were removed from further testing. The training ended when each of the associations was correctly recalled once. On average, each association was tested 2.3 times (SD = 0.26). On Day 2, subjects were again instructed to recall A-B associations. The procedure was identical to A-B overtraining on Day 1. On average, each association was tested 2.03 times (SD = 0.24). The purpose of the A-B over-training was to ensure that subjects had strong A-B memory.

### A-C updating

Fifteen minutes after the A-B over-training on Day 2, subjects were then introduced to the new A-C associations outside the scanner and were asked to replace the A-B associations with the new A-C associations. The procedure was identical to A-B learning except that the pictures associated with word cues were changed to pictures from a different category. Subjects were asked to study and remember the new A-C associations, and all old A-B associations would be irrelevant to any future tasks. The specific instructions were as follows: 'Now you will study some new picture-word associates. The words are all from the previously studied associations, but the paired pictures are all new. Your task is to remember these new associations. The old associations are irrelevant to any future task, so you do not need to remember them'. Subjects studied the new A-C memory once outside the scanner, and five minutes after initial A-C updating, they were put into the scanner to finish additional A-C updating. During scanning, half of the A-C pairs were assigned to the RetPrac condition and the other half to the Restudy condition. Under the RetPrac condition, a word cue and a black rectangle were presented for 2 s and subjects were asked to recall the picture associated with the presented (cue) word as vividly as possible. A red rectangle was then shown for 1 s and subjects were asked to judge the category of picture C by pressing a button corresponding to the visual category of the picture within this 1 s response window. The button-category correspondence was shuffled across subjects and only presented on the screen during the response stage (i.e., when the red rectangle was shown). This was to prevent subjects from planning motor response during the recall stage. After the response window, the correct picture was presented on the screen for 1 s. The next trial started after a jittered fixation (ranging from 0.5 s to 6.5 s, mean = 2 s). Under the Restudy

condition, the procedure was similar to that for the RetPrac condition except that the correct picture was presented on the screen during the entire trial. Subjects were still required to judge the category of picture C when the red rectangle was shown. The RetPrac and Restudy trials were pseudo-randomly intermixed within a run. Each trial was repeated three times, with an inter-repetition-interval (IRI) ranging from 10 to 36 (mean = 21.93). The IRI was matched between the two conditions. Each run of A-C updating contained 12 unique RetPrac pairs and 12 unique Restudy pairs and lasted 7.2 min. Subjects finished 6 runs of the learning task.

### A-C final test
On Day 3, subjects were instructed to perform the A-C final test while being scanned. A slow event-related design (12 s for each trial) was used to obtain a better estimation of the activation pattern for each item. After a 1 s fixation, a cue word and a black rectangle were presented on the screen for 4 s, and subjects were instructed to recall the picture C associated with the cue word as vividly as possible. The rectangle then turned red for 1 s and subjects were required to press a button to indicate the category of the retrieved picture. Similar to the RetPrac condition on Day 2, we also introduced the response label to prevent motor planning during the recall stage. Subjects were then asked to perform a Gabor orientation judgment task for 6 s. During this task, a Gabor pointing 45 degrees either to the left or right was presented and subjects were asked to judge the orientation of the Gabor as fast as possible. A self-paced procedure was used to make the task engaging. The A-C final test consisted of 3 runs of 9.6 min, each containing 48 associations.

### Follow-up behavior experiments (Exps 2 and 3)
Exp. 2 was designed to examine how new A-C memory practices would affect the old A-B memory. The procedure for Exp. 2 was nearly identical to that used for the main fMRI experiment, except that both A-C and A-B memories were tested and there was no perceptual orientation judgment task between retrieval trials. To examine the effect of test order, half of the subjects did the A-B memory test first and the other half did the A-C memory test first.

Exp. 3 was conducted to further examine whether RetPrac could effectively modify the details of memory representations. In this experiment, the procedure was identical to that used in Exp. 2, except that during the Day 3 test, we asked the subjects to write down the name of the associated picture (or any associated details if they could not recall the exact name) for each cue word, instead of choosing one of the three categories by pressing a button. They were first asked to write down the new C memory for each cue word A. After the A-C memory test, they were then asked to write the old B memory for each cue word. Only items for which subjects recalled the correct picture name or provided specific details were coded as correct items.

### Functional localizer
After the A-C final test, subjects were instructed to complete four runs of a functional localizer task, which was used to train the pattern classifier (see the multi-voxel pattern analysis section below for details). A mini block design was used in the task. Each run consisted of nine mini blocks of pictures from one of the three categories (three mini blocks per category). Within each mini block, six new word-picture associations were presented sequentially for 24 s, and subjects were asked to memorize these new associations. This procedure was used to match the perceptual and cognitive structure of the main task. The words used in this task were different from those used in the A-B or A-C pairs. The order of the mini blocks was counterbalanced across runs and subjects. After each mini block, there was a 12 s Gabor orientation judgment task using the self-paced procedure as described above.

## MRI acquisition
MRI data were collected on a 3.0T Siemens Prisma scanner (Siemens, Erlangen, Germany) with a 64-channel head coil at the Center for MRI Research at Peking University. A high-resolution simultaneous multi-slice EPI sequence was used for functional scanning (TR/TE/$\theta$ = 2000 ms/30 ms/81°; FOV = 220 × 220 mm; matrix = 116 × 116; slice thickness = 1.9 mm; GRAPPA factor = 2, multiband acceleration factor = 2; phase partial Fourier = 7/8). Seventy-two contiguous axial slices parallel to the AC-PC line were obtained to cover the whole cerebrum and cerebellum. A high-resolution

structural image using a 3D T1-weighted MPRAGE sequence was acquired to cover the whole brain (TR/TE/θ = 2530/2.98 ms/7°; FOV = 256 × 256 mm; matrix = 256 × 256; slice thickness = 1 mm; GRAPPA factor = 2).

## Image preprocessing

Image preprocessing and statistical analysis were performed using FEAT (FMRI Expert Analysis Tool) version 6.00, part of the FSL (FMRIB software library, version 5.0.9, www.fmrib.ox.ac.uk/fsl, RRID: SCR_002823) (*Smith et al., 2004*). The first 10 volumes before the task were automatically discarded by the scanner to allow for T1 equilibrium. The remaining images were then realigned to correct for head movements. Volumes with frame-wise displacement (FD) greater than 0.9 mm were discarded from further analysis. Data were spatially smoothed using a 5 mm FWHM Gaussian kernel and filtered in the temporal domain using a nonlinear high-pass filter with a 100 s cutoff. The EPI images were first registered to the MPRAGE structural image using affine transformation from FLIRT. Registration from structural image to the standard space was carried out using Advanced Normalization Tools nonlinear registration SyN (*Avants et al., 2011*). The transformation parameters from the two steps were combined into a single transform matrix in order to avoid multiple interpolations during EPI to standard space transformation.

For MVPA, fMRI data were preprocessed in the same way as for the univariate analysis except for spatial normalization. All preprocessed EPI volumes were registered to the first volume of the first A-C learning run using Advanced Normalization Tools (ANTs, RRID:SCR_004757) (*Avants et al., 2011*) and all MVPA were conducted in subjects' native EPI space.

## Definition of ROIs

We focused our MVPA on the ventral temporal cortex (VTC), medial prefrontal cortex (MPFC), angular gyrus (AG), and hippocampus (HPC), which overlap with the core recollection network (*Rugg and Vilberg, 2013*) and have consistently shown neural reinstatement effect during memory retrieval (*Kuhl and Chun, 2014*; *Wimber et al., 2015*; *Xiao et al., 2017*). The VTC and HPC were defined using the Harvard-Oxford probabilistic atlas (threshold at 25% probability). The VTC consisted of temporal fusiform, parahippocampus, and inferior temporal gyrus. The MPFC was defined on the basis of Automated Anatomical Labeling v2 and contained medial superior frontal gyrus, anterior cingulate gyrus, medial orbital superior frontal gyrus, and rectus gyrus (*Rolls et al., 2015*; *Figure 3A*). The AG was defined on the basis of the Schaefer2018 atlas (400-parcels) (*Schaefer et al., 2018*) and contained all parietal nodes within the default mode network (DMN, Network 15–17). All ROIs contained brain regions from both hemispheres.

## Univariate analysis

The general linear model within the FILM module of FSL was used to model the data. Two separate models were specified for the encoding phase. The first GLM aimed to compare the neural activation under the RetPrac and Restudy conditions during A-C updating. In this model, each repetition for each condition was separately modeled. The 'don't know' trials and the 'no response' trials from both conditions and all repetitions were separately modeled as two regressors of no interest.

The second model aimed to further examine the neural activations associated with updating performance in the RetPrac condition. Three trial types were defined on the basis of the updating history: trials in which a given item was correctly recalled the first time the pair was shown (first correct, FC), trials in which a given item was correctly recalled the second or third time the pair was shown (later correct: LC), and incorrect trials (IC). In addition, items that were correctly recalled first time but that were followed by incorrect responses in later repetitions, which were very rare, were separately modeled as two regressors (correct, incorrect) of no interest. In addition, each of the three repetitions in the Restudy condition, the 'don't know' and 'no response' trials were also separately modeled.

All regressors were convolved with a double gamma hemodynamic response function (HRF). Their temporal derivatives were also included. Each run was modeled separately in the first-level analysis. Using a fixed-effects model, cross-run averages for a set of contrast images were created for each subject. Each contrast image for all subjects was entered into group analysis using the non-parametric permutation method for inference on the statistic map. This was conducted by the Randomise

program in FSL with 10,000 permutations. The significance of the derived statistical map was determined by the threshold-free cluster enhancement (TFCE) algorithm with p<0.05 (whole brain FWE corrected) (*Smith et al., 2004*).

In order to estimate the neural activation for each item across the three repetitions, we modeled each item's three repetitions as a separate regressor in one GLM. The parameters estimated from this procedure served as an averaged activation measure across the three repetitions of a given association. The analysis was conducted at native space and the statistical maps were transformed to the MNI space. The activations in the ROIs were then used for the linear mixed-effect model.

## Multi-voxel pattern analysis

Neural reactivation was quantified by the classifier output from a pattern classifier trained on separate functional localizer data. The trained classifier was used on both updating and testing data to assess the level of reactivation of categorical information. All MVPA were conducted on each subject's native anatomical space. The data were normalized with the following steps based on a previous study (*Kuhl et al., 2012*). The data were first z-scored within a scan, and then across voxels within each volume. After all relevant volumes had been selected, data were z-scored again across all updating/final test/localizer volumes. Because each trial in the updating/final test/localizer phase corresponded to multiple fMRI volumes, the data were averaged across volumes within a trial before pattern analysis. For the testing stage, a weighted average was performed across 3–6 TRs after cue word onset (corresponding to 4–12 s after cue word onset, weights = [0.35, 0.35, 0.15, 0.15]). For updating and localizer trials, data were averaged across 3–4 TRs after cue word/picture onset. The choice of time window was based on a previous study (*Kuhl et al., 2012*) with consideration of hemodynamic lag.

Classification analysis was performed using L2-norm regularization logistic regression with liblinear solver from scikit-learn package (RRID:SCR_002577) in Python. Three binary classifiers (one vs the rest) were trained on functional localizer data within each pre-defined ROI (*Fan et al., 2008*). Individual classifier's output probabilities were averaged together on the basis of category to form the final output probability. The penalty parameter C was set to 0.01 following a previous study (*Kuhl et al., 2012*). Each picture category had 72 samples (trials) for classifier training. To examine the performance of the classifier, we applied the leave-one-run-out cross-validation procedure. The average cross-validation accuracy of the classifiers was significantly above the chance level (ranging from 51.0% to 83.9%, all p-values <0.001; *Figure 3—figure supplement 1A*).

Then, the classifier trained on all functional localizer runs was applied to both updating and testing data. Classifier output probability for each category (face, object, or scene) was assigned to target, competitor, or others, based on the categories of pictures B and C. Specifically, as each cue word A was associated with two pictures, and the goal was to memorize A-C associations and to inhibit A-B associations, so the classifier output corresponding to categories of the C and B images was assigned as the target and competitor outputs, respectively, whereas the classifier output corresponding to the remaining category was assigned to the 'other' output. The classifier evidence was normalized within each trial by subtracting the 'other' evidence from the target and competitor evidence.

## Nonmonotonic plasticity curve fitting

A Bayesian curve-fitting procedure was used to fit the nonlinear relationship between competitor reactivation strength during the updating phase and the relative competitor reactivation strength change during the final test. This analysis was performed using the p-cit-toolbox with its default parameter settings (*Detre et al., 2013*). In brief, the algorithm approximates the most probable plasticity curve from the given data (competitor reactivation during updating and the final test). First, it generates the curve by sampling possible curves (linear curves with three segments) randomly. Then, an importance weight is given by how well the curve fits the actual data. Finally, the mean curve is generated by determining the weighted average of all the sampled curves. In addition, the fitted curves were divided into two groups: theory-consistent (a U-shaped curve) or theory-inconsistent. The P(theory-consistent) is computed as the fraction of posterior probability of the theory-consistent curve samples. This value indicates how well the data support the nonmonotonic curve. Trials under the RetPrac and the Restudy conditions were modeled separately for each ROI. A permutation

test (1000 permutations) was used to determine the significance of the posterior probability of theory consistency [P(theory consistent)].

## Statistical analysis

All repeated measures ANOVAs were conducted in the afex package using Type III sums of squares in R 3.3.3 (RRID:SCR_001905). The error bars in the figures denote within-subject errors that account for heterogeneity of variance. FDR correction ($\alpha$ = 0.05) was performed to correct for post-hoc multiple comparisons. We reported the uncorrected p-values in the main text and indicated whether they were significant with FDR correction. Cohen's d was calculated as a measure of effect size for main comparisons.

## Mixed-effects modeling

Mixed-effects modeling is a powerful statistical tool that offers many advantages over conventional t test, regression, and ANOVA in sophisticated fMRI designs. The linear mixed-effects model used in this study was implemented with lme4 in R (RRID:SCR_015654), fitted using restricted maximum likelihood. To determine the effect of the predictor of interest, we used the likelihood ratio test to compare models with (full model) and without (null model) predictor of interest. For the caudate/LPFC activation and competitor suppression model, FC or FC+IC trials' activation for a given ROI was used as the predictor, and the difference in competitor evidence between the current repetition and subsequent repetition during updating was used as the dependent variable. For the caudate/LPFC activation and long-term memory updating model, a given ROI's activation for all three repetitions were averaged as the predictor, and the target minus competitor evidence during the final test was used as the dependent variable. For all models reported in the main text, the random intercept was included as a random effect.

## Acknowledgements

We thank Michael Anderson for constructive comments on the manuscript. This work was sponsored by the National Science Foundation of China (31730038), the NSFC and the Israel Science Foundation (ISF) joint project (31861143040), the NSFC and the German Research Foundation (DFG) joint project NSFC 61621136008/DFG TRR-169, and the Guangdong Pearl River Talents Plan Innovative and Entrepreneurial Team grant #2016ZT06S220.

## Additional information

### Funding

| Funder | Grant reference number | Author |
|---|---|---|
| National Science Foundation of China | 31730038 | Gui Xue |
| The NSFC and the Israel Science Foundation (ISF) joint project | 31861143040 | Gui Xue |
| National Science Foundation of China | 61621136008 | Gui Xue |
| German Research Foundation | TRR-169 | Gui Xue |
| Guangdong Pearl River Talents Plan Innovative and Entrepreneurial Team grant | 2016ZT06S220 | Gui Xue |

The funders had no role in study design, data collection and interpretation, or the decision to submit the work for publication.

### Author contributions

Zhifang Ye, Conceptualization, Resources, Data curation, Software, Formal analysis, Investigation, Visualization, Methodology, Writing - original draft, Writing - review and editing; Liang Shi, Anqi Li,

Investigation; Chuansheng Chen, Writing - review and editing; Gui Xue, Conceptualization, Resources, Supervision, Funding acquisition, Methodology, Writing - original draft, Writing - review and editing

**Author ORCIDs**
Zhifang Ye (iD) https://orcid.org/0000-0003-0489-2619
Gui Xue (iD) https://orcid.org/0000-0001-7891-8151

**Ethics**
Human subjects: Written consent was obtained from each subject after a full explanation of the study procedure. The study was approved by the Institutional Review Boards at Beijing Normal University and the Center for MRI Research at Peking University (#20150401).

**Decision letter and Author response**
Decision letter https://doi.org/10.7554/eLife.57023.sa1
Author response https://doi.org/10.7554/eLife.57023.sa2

# Additional files

### Supplementary files
• Supplementary file 1. Update method (RetPrac, Restudy) x Test order (A-C first, A-B first) two-way ANOVA table by Memory test type and Response type.

• Supplementary file 2. ANOVA tables of classifier evidence tests. (a) Updating method (RetPrac, Restudy) x Classifier evidence (Target, Competitor) x Memory outcome (Correct, Incorrect) three-way ANOVA table. (b) Updating method (RetPrac, Restudy) x Memory outcome (Correct, Incorrect) two-way ANOVA table by Classifier evidence type.

• Supplementary file 3. Tables of univariate activation contrasts. (a) Regions showing greater activation during updating under the RetPrac condition than under the Restudy condition. (b) Regions showing greater activation during updating under the Restudy condition than under the RetPrac condition. (c) Pairwise comparisons of brain activity between trials with different updating performance.

• Transparent reporting form

### Data availability
All fMRI data collected in this study is available on OpenNeuro under the accession number 002773 (https://openneuro.org/datasets/ds002773).

The following dataset was generated:

| Author(s) | Year | Dataset title | Dataset URL | Database and Identifier |
|---|---|---|---|---|
| Ye Z, Xue G | 2020 | Retrieval practice facilitates memory updating by enhancing and differentiating medial prefrontal cortex representations | https://doi.org/10.18112/openneuro.ds002773.v1.0.0 | OpenNeuro, 10.18112/openneuro.ds002773.v1.0.0 |

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
