## [Decision Letter]

**Acceptance summary:**

This elegant human study examined the effects of retrieval practice on memory performance and neural responses. The results from a set of experiments show that retrieval practice strengthens new memories and reduces intrusions of old memories without suppressing the old memories. Interestingly, this was related to enhanced representations in medial prefrontal cortex, further supporting the idea that this region is important for memory integration and consolidation.

**Decision letter after peer review:**

[Editors’ note: the authors submitted for reconsideration following the decision after peer review. What follows is the decision letter after the first round of review.]

Thank you for submitting your work entitled "Retrieval practice facilitates reactivation-dependent memory updating" for consideration by *eLife*. Your article has been reviewed by three peer reviewers, and the evaluation has been overseen by a Reviewing Editor and a Senior Editor. The reviewers have opted to remain anonymous.

Our decision has been reached after consultation between the reviewers. Based on these discussions and the individual reviews below, we regret to inform you that your work will not be considered further for publication in *eLife*.

As is evident in the individual critiques copied below, the reviewers agreed that your manuscript provides nice and robust evidence for memory updating by means of strengthening the target memories. However, the reviewers felt that given the lack of strong evidence for either competitor suppression or integration, this alone is not novel enough for *eLife*. In addition, the reviewers agreed that results from the 3-way classifier are difficult to interpret and that two separate 2-way classifiers or an RSA approach would be required to reveal what is reactivated. Finally, there were concerns that the results of experiment 2 were inconclusive.

In light of these concerns, we decided that we cannot move forward with this paper at *eLife*. However, if you are willing and able to address these issues in full (including new behavioral experiments), you would be able to resubmit a substantially revised version of the manuscript to *eLife* in the future (if you choose to do this, please refer to this manuscript number and rejection decision in your future submission). However, we understand that you may prefer to submit the manuscript, in its current form, to a more specialized journal.

Reviewer #1:

The authors test the hypothesis that active remembering is an effective means of memory updating when there is proactive interference from outdated information. Participants learn novel associations (A-B) on Day 1, then are asked to replace these memories with overlapping associations (A-C) on Day 2, either by active retrieval practice (RetPrac) or Restudy practice of the new A-C memories. On Day 3 all groups undergo a final memory test. Behaviorally, participants in the RetPrac condition recall the target more often, and experience fewer competitor intrusions, on the Day 3 final test. Multivariate analyses of brain activity patterns suggest that competitors are co-activated on Day 2 in a number of brain regions, but subsequently suppressed and become less accessible on Day 3, again specifically in the RetPrac items. Neural evidence for target and competitor reactivation thus largely tracks the behavioral effects observed.

The manuscript addresses an interesting and timely topic, and the Introduction and Discussion are well written and accessible. The most central behavioral and MVPA findings seem very robust. These findings are largely to be expected based on the existing literature (as appropriately cited in the manuscript), but I do not know of a similar set of coherent results that demonstrates these aspects of memory updating, and therefore I think the manuscript should in principle be considered for publication in *eLife*. I have a few major concerns, however, regarding the methods, results and conclusions, as outlined in the following.

1) A fundamental technical problem in my view is the use of a 3-class classifier. When such a classifier provides evidence for a target, it is automatically confounded by "anti-evidence" for the competitor and the other, neutral category. The same goes for the competitors: if a 3-class classifier provides strong evidence for competitors being reactivated, this could be due to strong competitor evidence or weak target (and neutral) evidence. The problem is exaggerated by using a normalization where the "neutral" evidence is subtracted from target and competitor evidence. It would be much preferable if the authors used 2-class classifiers with the neutral category as baseline for all analyses trying to separate target and competitor reactivation.

2) During retrieval practice, were subjects significantly more likely to experience an intrusion than an unrelated (other) error? It seems from Figure 1C that they were not, speaking against any strong A-B proactive interference effect, and more for a general learning of the target response over time. This result seems crucial for the central claims of the paper. Similarly, the authors should report in both experiments how the number of Day 2 intrusions on relates to Day 3 performance, and whether this relationship is stronger for intrusions than unrelated errors.

3) The authors make strong claims about competitor suppression throughout the manuscript where the actual evidence appears relatively weak. In the behavioural Experiment 2, if there was a suppression effect, how do the authors explain that they did not find reduced A-B memory for RetPrac than Restudy on Day 3? Is this due to the delay of the final test? Relatedly, the negative correlation between A-B and A-C memory is interpreted as evidence for suppression but can as easily be seen as an associative interference effect. Finally, in Experiment 2, the authors test A-C associations before A-B associations, making it likely that output interference on the A-B items will overshadow potentially more subtle effects of suppression in this experiment.

4) While the most central results seem sound, some of the analyses reported later in the Results section appear less well motivated and somewhat arbitrary in their approaches and selective in the reporting. To avoid the impression of p-hacking, the authors should streamline these sections, and use a more consistent and well-motivated rationale for all of the analyses. To give a few concrete examples,

a) The analysis reported in paragraph two of subsection “The LPFC contributed to memory updating under the RetPrac condition” and in Figure S8 is difficult to follow in terms of rationale. The results are also a bit confusing: none of the ROIs shows a pattern where strong competitor reactivation is related to strong LPFC activation, which is surprising given existing literature. Did the authors average competitor evidence across the 3 repetitions? In my mind, the straightforward prediction here is be that strong competitor reactivation on early repetitions, and weak on late repetitions, should be related to effective LPFC-mediated suppression. Therefore, for these analyses it makes more sense to use the slope of competitor reactivation across repetitions, not the average evidence for reactivation

b) The analysis relating caudate activity to competition resolution seems arbitrary for readers not reading the supplements. It is unclear why a different metric is being used here (compared to LPFC) to relate univariate and multivariate effects.

c) The analysis splitting trial into incorrect (IC), first correct (FC) and second/third correct (LC) is not well motivated and difficult to follow.

5) In some instances, interpretations are quite a large step removed from the actual results. For example, why does a difference in LPFC activity between IC/FC and LC (paragraph two subsection “The LPFC contributed to memory updating under the RetPrac condition”) indicate a role in competitor suppression? Such a pattern is more likely driven by target-related processes.

6) The tertile analyses relating competitor reactivation on Day 2 to competitor reactivation (see Figure 5) on Day 3 are not convincing statistically. The interaction with region as a factor seems irrelevant. In IFG and AG the conclusions are based on null results, and in VTC there is a strong positive relationship speaking against reactivation-dependent updating. The only thing left therefore is the U-shaped effect, and this appears like a posthoc observation.

7) For the Day 2 MVPA analyses, the authors never show evidence for target reactivation (or rather, representation given the visual exposure), this result should be included.

Reviewer #2:

In this timely and creative study, the authors investigate in a within-subjects fMRI design with 19 subjects the neural mechanisms and behavioral effects of retrieval practice vs. re-study of A-B, A-C memory updating. In a second within-subjects behavioral study with 28 subjects, the authors probe their postulation that retrieval practice vs. re-study leads to prioritization of C and suppression of B as related to A in memory updating. There are a few drawbacks to the theoretical framing, analytic technique, and conclusions drawn from the data that should be addressed.

1) Theoretical framing:

The authors adopt a research paradigm that is akin to inference/integration/generalization memory work. I was surprised to see this conceptualization of the paradigm downplayed in the Introduction and Discussion, particularly because A-B pairs were overtrained on Day 1. I think that the manuscript would benefit from more explicit characterization of why the adopted paradigm speaks to a suppression account of a B vs. an integration account of a B in memory updating, and how their data adjudicate between these two accounts (e.g., MPFC is often shown as a schema/generalization area).

2) Analytic technique:

a) I was not convinced that the retrieval practice vs. re-study Day 2 design and analyses provided clear support for the idea that B is suppressed and C is prioritized in retrieval practice vs. re-study during memory updating. In fact, the authors do not find evidence of this claim in their follow-up behavioral study designed to address the issue; they have to split data to show that in some subjects you see this pattern but in others you don't. Because the evidence stemming from this follow-up is not clear, the mechanism of retrieval practice in memory updating (suppression vs. integration) is not clear.

b) The critical results of the study are subsection “Retrieval practice enhanced target reactivation and competitor suppression” of the manuscript. I was confused as to why the authors have one classifier analysis for final test performance (Day 3) that doesn't account for retrieval practice/re-study (Day 2), and then another classifier analysis for retrieval practice/re-study (Day 2) that doesn't account for final test performance (Day 3). Why was behavioral performance but not the Day 2 manipulation used in the first analysis, and why was the Day 2 manipulation but not behavioral performance used in the second analysis? I think an analysis that uses both these assays would get at the authors' question most directly.

c) How many trials were used in the fMRI analyses per condition?

d) Greater motivation for the ROI selection parameters for MVPA should be given.

e) Is it possible that shifts in decision criteria would be observed for those trials with retrieval practice on Day 2 vs. those trials with re-study on Day 2? Can the authors rule out a decision criteria account of behavioral findings on Day 3 final test?

3) Conclusions drawn from the data:

a) As noted, I did not think that the critical finding about retrieval practice during memory updating was supported by the data; the authors' own follow-up study did not provide strong behavioral support for this claim. Thus, it is hard to reconcile the lack of a clear mechanism with the fMRI side of the manuscript.

b) The lack of neuroimaging effects for AG and Hipp was striking, particularly given recent work finding reactivation effects in parietal cortex (Jonker, Ranganath, 2019, PNAS; Lee, Kuhl, 2019, Cerebral Cortex). These papers should be cited in text and discussion should be given as to the discrepancies.

Reviewer #3:

In this paper, the authors tested the hypothesis that retrieval facilitates memory updating through stronger suppression of competing memory. Using a word-picture association paradigm with fMRI, the authors found that brain regions, including mPFC, showed greater reactivation of new memory at the final test, but greater reactivation of old memory during practice in the test condition, compare with the restudy condition. In addition, LPFC showed stronger activity during retrieval practice than during restudy. Overall, this is a very interesting study addressing how memory retrieval interact with proactive memory interference. The paper is easy to read and the design and method are clearly described. However, some of our enthusiasm was dampened by significant questions and concerns regarding the novelty and central arguments in the paper. These areas of significant concern are detailed below:

1) This is a very interesting and fruitful set of results, but the framing does not seem to fit with the work that is reported. The logic of the paper is that neural changes in response to repeated retrievals reflect memory updating and suppression of old memory, but a priori this did not seem like an obvious prediction. It is possible that any differences (both in behavioral performance and neural activity) between the test and the restudy condition simply reflect superior learning during testing. In other words, without a no interference or weaker interference condition, it is hard to conclude that brain activity during retrieval practice supports memory updating/suppression.

2) From the Introduction, it is unclear how this study is different from prior studies examining neural mechanisms underlying retrieval induced forgetting using a similar paradigm, e.g. Wimber et al., 2015.

3) The authors capture many previous findings about activity in mPFC and LPFC, however, the hypothesis they form following this literature is too vague to be clearly falsifiable or to adjudicate between potentially contradictory findings. For instance, rather than reflecting suppression, stronger reactivation of competing memory during retrieval practice has also been associated with retrieval induced facilitation (e.g. Jonker et al., 2018). Moreover, the authors examine brain activity in 4 ROIs but did not present a rationale for including AG, IFG, VTC.

4) It is laudable the authors report a follow-up behavioral experiment examining the relationship between memory for new memory vs. old memory. However, the negative correlation could be driven by output interference, especially given that subjects recalled A-C pairs first. It is likely that better A-C recall produced stronger output interference to A-B pairs.

5) It is unclear in the analyses of retrieval practice reactivation, whether the authors included all test trials or only correct test trials. If all trials were included, all the results reported in these sections would not be surprising because subjects wrongly recalled a large portion old targets during retrieval practice. This also explains why restudy trials showed larger reactivation of new memory because new information was always directly presented.

6) A number of ad hoc hypotheses are given for results that are inconsistent with the prediction. For example, the authors claim that the null results for correct or incorrect trials are due to a small number of trials. However, there should be at lease 70 correct trials in the test condition and 50 correct trials in the restudy condition. Moreover, decreased update in the restudy condition is explained by the repetition suppression effect; null results of correlations between competitor reactivation and behavioral performance are explained by the claim that behavioral measure is not sensitive; chance level classification performance in the hippocampus is thought to reflect "technical limitations". These ad hoc explanations of conflicting results, which lack justification, suggest strong confirmation bias.

[Editors’ note: further revisions were suggested prior to acceptance, as described below.]

Thank you for submitting your article "Retrieval practice facilitates memory updating by enhancing and differentiating medial prefrontal cortex representations" for consideration by *eLife*. Your article has been reviewed by two peer reviewers, and the evaluation has been overseen by a Reviewing Editor and Laura Colgin as the Senior Editor. The reviewers have opted to remain anonymous.

The reviewers have discussed the reviews with one another and the Reviewing Editor has drafted this decision to help you prepare a revised submission.

Summary:

All reviewers agreed that the authors have done a very thorough job addressing the initial comments, and that, as a result, the paper is much improved. Reviewers also identified a few remaining issues that should be addressed with new analysis and re-writing.

Revisions:

1) The authors analyzed the behavioral data with a "test order (A-C first, A-B first) by memory test type (Recall A-C, Recall A-B) by update method (RetPrac, Restudy) by response type (Target, Competitor, Other) four-way mixed design ANOVA". Either the choice or the description of the analysis is incorrect. Given that the performance of A-C test and A-B test, absolute ratios of Target, Competitor and Others are not comparable, response type and memory test type should not be used as independent variables (factors). Rather, separate ANOVAs should be conducted, with appropriate multiple comparison correction, to examine target ratio and competitor ratio in each test.

2) Given the above reason, the authors cannot rule out the potential order confounding by just showing there was no significant effect of test order or interaction effect with test order in this four-way ANOVA. Rather, the authors need to directly compare A-B performance between when A-B was tested first vs. when A-B was tested after A-C. However, even if the correct analysis was done, non-significant difference between different orders cannot rule out the confounding of order either. A stronger test would be only examining A-B performance with subjects start with A-B test, and vice versa for A-C test.

---

## [Author Response]

[Editors’ note: the authors resubmitted a revised version of the paper for consideration. What follows is the authors’ response to the first round of review.]

Reviewer #1:[…]1) A fundamental technical problem in my view is the use of a 3-class classifier. When such a classifier provides evidence for a target, it is automatically confounded by "anti-evidence" for the competitor and the other, neutral category. The same goes for the competitors: if a 3-class classifier provides strong evidence for competitors being reactivated, this could be due to strong competitor evidence or weak target (and neutral) evidence. The problem is exaggerated by using a normalization where the "neutral" evidence is subtracted from target and competitor evidence. It would be much preferable if the authors used 2-class classifiers with the neutral category as baseline for all analyses trying to separate target and competitor reactivation.

We thank this reviewer for the comment. In this study, instead of just using a target (T) and a competitor (C), the third neutral (other, O) condition was included exactly to deal with the anti-evidence issue. As a result, we could simultaneously detect strong target and competitor evidence.

As correctly pointed by this reviewer, the solver “liblinear” used a “one-vs-rest” fashion to do multi-class classification (i.e., T vs. O+C, C vs. T+O and O vs. T+C), which is a common practice in machine learning field et al.(Fan , 2008). Other studies have used three binary classifiers (i.e., T vs. C, T vs. O, and O vs. C) and combined the results from the classifiers. For example, the evidence for T would be obtained by averaging the evidence from classifier T vs. C and T vs. O. The reviewer here suggested that we did not use the evidence from the classifier T vs. C, since this would introduce the anti-evidence issue.

We very much appreciate this reviewer’s point. Nevertheless, it should be noted that with the classifier, we were to assess the relative strength of evidence from each class, but not so much about the absolute value. In addition, both 3-class and binary classifier approaches could maximize the number of training samples for each classifier, which potentially leads to more stable results.

In light of the reviewer’s comment, we additionally implemented the one-against-one classifiers as a comparison. Specifically, we implemented 3 binary classifiers, each using data from two categories. The target and competitor evidence was obtained from the T vs. O and the C vs. O classifier, respectively, without using the results from the T vs. C classifier. The Other evidence was the average from the two classifiers. The results from this analysis were overall very similar to our results in the manuscript (see Author response image 1).

Since the one-against-the-rest method is a common practice in the machine learning field and the three-classifier approach might be robust and consistent with the literature, we decided to use the results from three-class classification. In in the Materials and methods section of the revised manuscript, we have added some rationale for the use of this approach.

2) During retrieval practice, were subjects significantly more likely to experience an intrusion than an unrelated (other) error? It seems fromFigure1C that they were not, speaking against any strong A-B proactive interference effect, and more for a general learning of the target response over time. This result seems crucial for the central claims of the paper. Similarly, the authors should report in both experiments how the number of Day 2 intrusions on relates to Day 3 performance, and whether this relationship is stronger for intrusions than unrelated errors.

This is a very good point. In all of our three experiments, we did observe slightly more competitor intrusions than unrelated errors (Experiment 1: 0.232 vs. 0.209, t_(18)_ = 1.93, p = .069; Experiment 2: 0.233 vs. 0.207, t_(45)_ = 2.72, p = .009; Experiment 3: 0.221 vs. 0.200, t_(27)_ = 2.33, p = .027). We think this could reflect the fact that subjects were told explicitly that the associations had been changed, and new associations should be explored and learnt. That was why we also found many other responses. Still, there were stronger neural competitor reactivations than other evidence during memory updating. Finally, the competitor response was much stronger during the final test on Day 3. All these pieces of evidence suggest strong A-B proactive interference.

Following this reviewer suggestion, we did additional analyses to examine how the number of Day 2 intrusions was related to Day 3 performance. As shown in Figure 2—figure supplement 1, we found that pairs with more competitor intrusions during memory updating had worse A-C memory, stronger A-B intrusion, and comparable other response during final A-C memory test. This pattern was consistent in both Experiment 1 and Experiment 2. In contrast, pairs with more other response during memory updating had worse A-C memory, stronger other response, but comparable A-B intrusion, during the final A-C memory test. These results suggested that the strength of proactive interference affected the new A-C learning. Importantly, although the number of competitors and other responses had a similar effect on A-C memory, they had a much stronger effect on the accessibility of themselves during the final test. That is, there were much more A-B intrusions than other responses during the final test, even when the number of response during updating was matched, suggesting strong A-B interference.

Interestingly, we found that the number of competitor and other responses during updating had no effect during the A-B memory test, suggesting that the consolidated A-B memory was not weakened.

Together, these results suggest that there was very strong A-B proactive interference both during updating and the final test. It had a strong effect on the learning of new memory, but not on the strength of A-B memory itself, which is consistent with the differentiation hypothesis. We have added these results in the manuscript, which reads:

“Could the lack of suppression be due to the weak A-B intrusion? Subjects were well trained in the A-B association. […] All these results suggested that the lack of A-B memory suppression was due to its strong representations.”

3) The authors make strong claims about competitor suppression throughout the manuscript where the actual evidence appears relatively weak. In the behavioural Experiment 2, if there was a suppression effect, how do the authors explain that they did not find reduced A-B memory for RetPrac than Restudy on Day 3? Is this due to the delay of the final test? Relatedly, the negative correlation between A-B and A-C memory is interpreted as evidence for suppression but can as easily be seen as an associative interference effect. Finally, in Experiment 2, the authors test A-C associations before A-B associations, making it likely that output interference on the A-B items will overshadow potentially more subtle effects of suppression in this experiment.

All three reviewers raised this important issue as to whether our data reflected competitor suppression. In this revision, we did an additional behavioral experiment to examine the A-B memory in Day 3. Two groups of subjects (23 in each group) were included, with half tested A-B memory first and the other half tested A-C memory first. Our results replicated the RetPrac effect on A-C memory, but again revealed no effect on A-B memory. This was the case for both groups, suggesting that the order of the tests did not have an effect.

The results from the new experiment, together with those from the original Experiment 2, support the differentiation mechanism. That is, although A-C memory was strengthened by retrieval practice, but there were not more C intrusions in the A-B memory test. On the other hand, although A-B memory was comparable between the two conditions, there were fewer B intrusions in the A-C memory test. Both sets of results suggest that A-B and A-C memories were differentiated by retrieval practice. We noticed that the number of C memory intrusions was overall low, perhaps due to the weak A-C memory. We thus did a further analysis focusing on strong A-C memory trials (correct trials in the A-C memory test), which would produce stronger intrusion. Consistently, we found that the correct trials in the A-C memory test (reflecting strong A-C memory) showed overall more C memory intrusions during the A-B test than did the incorrect trials (t_(45)_ = 3.38, p = .001). More interestingly, the intrusion rate was numerically smaller in the RetPrac than the Restudy condition (t_(45)_ = -1.59, p = .12), which is consistent with the differentiation hypothesis.

To further test the differentiation hypothesis, we did a joint analysis to examine the subjects’ answers in both A-B and A-C memory tests given the same word cues. The differentiation hypothesis would predict more correct trials in both tests (i.e., C response in the A-C memory test and B response in the A-B memory test) under the RetPrac condition, i.e., subjects maintained stronger and nonoverlapping representations of both A-B and A-C memories. Meanwhile, we would predict fewer trials on which subjects responded with old B memory in both tests (due to differentiation), and also fewer (due to differentiation) or comparable (due to strengthening of C and differentiation) trials on which subjects responded with new C memory in both memory tests. Our data supported all three predictions. We found that subjects made more correct responses in both tests under the RetPrac than the Restudy condition (24.0% vs. 19.7%, t_(45)_ = 3.81, p <.001), but showed less B memory (21.0% vs. 23.2%, t_(45)_ = -2.61, p = .012) and comparable C memory (11.7% vs. 11.5%, t_(45)_ = 0.18, p = .86) in both memory tests.

We also conducted new analyses on the fMRI reactivation data, by simultaneously examining the factors of updating method and behavioral performance. These results suggest memory integration and differentiation in the MPFC. In particular, retrieval practice was able to boost the target reactivation for both correct and incorrect trials. For competitor evidence, however, we found strong and comparable competitor reactivation under the RetPrac condition for both correct and incorrect responses, suggesting that RetPrac integrated and differentiated competitor and target evidence in the MPFC.

These new behavioral results, together with the neural representational data, suggest that reactivation during retrieval practice could enhance target representation, and meanwhile facilitate memory differentiation and reduce intrusion, which is very consistent with the recent hypothesis that emphasizes the role of reactivation in differentiating neural representations and (Ritvo, Turk-Browne Norman, 2019).

The reactivation-dependent memory differentiation is also quite consistent with previous behavioral observations and and(Keresztes Racsmány, 2013; Storm, Bjork Bjork, 2008). The lack of suppression might be due to several reasons. First, the old memory might be too strong to be suppressed. Second, there was a long delay between retrieval practice and the final memory test, as previous studies have found that strong retrieval induces suppression after a short delay, but much weaker effect after a 24-hour delay and(MacLeod Macrae, 2001).

In this revision, we have thoroughly rewritten the Introduction, Results and Discussion.

4) While the most central results seem sound, some of the analyses reported later in the Results section appear less well motivated and somewhat arbitrary in their approaches and selective in the reporting. To avoid the impression of p-hacking, the authors should streamline these sections, and use a more consistent and well-motivated rationale for all of the analyses. To give a few concrete examples,a) The analysis reported in paragraph two of subsection “The LPFC contributed to memory updating under the RetPrac condition” and in Figure S8 is difficult to follow in terms of rationale. The results are also a bit confusing: none of the ROIs shows a pattern where strong competitor reactivation is related to strong LPFC activation, which is surprising given existing literature. Did the authors average competitor evidence across the 3 repetitions? In my mind, the straightforward prediction here is be that strong competitor reactivation on early repetitions, and weak on late repetitions, should be related to effective LPFC-mediated suppression. Therefore, for these analyses it makes more sense to use the slope of competitor reactivation across repetitions, not the average evidence for reactivationb) The analysis relating caudate activity to competition resolution seems arbitrary for readers not reading the supplements. It is unclear why a different metric is being used here (compared to LPFC) to relate univariate and multivariate effects.c) The analysis splitting trial into incorrect (IC), first correct (FC) and second/third correct (LC) is not well motivated and difficult to follow.

We apologize for not making this clear in our original submission. The second/third correct (LC) indicates that items were correctly answered two or three times, indicating a relatively more fluent response. Our prediction was that the incorrect and first correct trials would involve more conflict resolution than would the LC trials. Furthermore, the first correct trials would involve more reward response than would incorrect and later correct trials. This way, we could dissociate the regions associated with reinforcement learning and reward processing (such as the caudate) and the regions involved in conflict processing (such as the prefrontal cortex). As suggested by et al.Ritvo (2019), both mechanisms could lead to representational changes and memory differentiation. We have clarified the motivation of this analysis in the revised manuscript, which reads:

“To further probe the function of these regions in overcoming intrusions from outdated competitors, we examined how these prefrontal and striatal activations varied with updating performance during retrieval practice. […] Both mechanisms could contribute to representational change and memory differentiation.”

For comments 4a and b, we agree with this reviewer that it would make more sense to link the caudate and LPFC activation with subsequent change of competitor evidence. In particular, we examined the caudate and LPFC activation in the current trial and the evidence change from the current repetition to next repetition. Due to caudate response to reward, we focused on the first correct trial (FC) in the caudate, which revealed strong caudate activation was associated with greater competitor evidence reduction in the next repetition in the VTC (χ^2(1)^ = 5.86, p = .015) but not in MPFC and AG (all ps >.15), suggesting reinforcement learning. However, no such effect was found for LPFC during the incorrect and first correct trials, suggesting the LPFC did not temporally suppress the competitor evidence.

We further examined whether the LPFC and caudate activations during updating were associated with long-term memory updating on Day 3. This analysis revealed that trials with greater LPFC activity during updating under the RetPrac condition ultimately showed superior memory updating (i.e., target – competitor evidence) during the final test on Day 3, in the MPFC (χ^2^_(1)_ = 4.62, p = .032), but no effect was found in the caudate.

Together, these results suggest different roles of LPFC and caudate in modifying the memory representations during retrieval practice. We have updated the results, and added more discussions in this revision.

5) In some instances, interpretations are quite a large step removed from the actual results. For example, why does a difference in LPFC activity between IC/FC and LC (paragraph two subsection “The LPFC contributed to memory updating under the RetPrac condition”) indicate a role in competitor suppression? Such a pattern is more likely driven by target-related processes.

As explained above, IC/FC trials were slower and required more conflict resolution, whereas LC trials were more fluent. We would expect similar responses for FC and LC trials in regions associated with target processing. We have clarified the logic of our interpretation in the revised manuscript, as stated above.

6) The tertile analyses relating competitor reactivation on Day 2 to competitor reactivation (see Figure 5) on Day 3 are not convincing statistically. The interaction with region as a factor seems irrelevant. In IFG and AG the conclusions are based on null results, and in VTC there is a strong positive relationship speaking against reactivation-dependent updating. The only thing left therefore is the U-shaped effect, and this appears like a posthoc observation.

We agree with this reviewer that the tertile analysis might not provide very strong evidence. In this revision, we used the more established method (Bayesian curve-fitting implemented by p-cit-toolbox) et al.(Detre , 2013) to formally test the relationship between reactivation during updating and memory evidence in the final test. We found a pattern that was consistent with the nonmonotonic plasticity hypothesis in the visual cortex, but not in other regions. We have reported the results from this new analysis.

7) For the Day 2 MVPA analyses, the authors never show evidence for target reactivation (or rather, representation given the visual exposure), this result should be included.

Thanks for this suggestion. In the revised submission, we reported the target evidence reactivation and related it to subsequent memory performance, separately for restudy and RetPrac conditions. We found that the target evidence during updating in the MPFC and VTC was separately associated with subsequent memory performance in the RetPrac and Restudy conditions, respectively, suggesting that RetPrac might involve the MPFC mechanisms for quick system consolidation. We did not directly compare the target evidence between the two conditions due to the differences in their task structure.

Reviewer #2:In this timely and creative study, the authors investigate in a within-subjects fMRI design with 19 subjects the neural mechanisms and behavioral effects of retrieval practice vs. re-study of A-B, A-C memory updating. In a second within-subjects behavioral study with 28 subjects, the authors probe their postulation that retrieval practice vs. re-study leads to prioritization of C and suppression of B as related to A in memory updating. There are a few drawbacks to the theoretical framing, analytic technique, and conclusions drawn from the data that should be addressed.1) Theoretical framing:The authors adopt a research paradigm that is akin to inference/integration/generalization memory work. I was surprised to see this conceptualization of the paradigm downplayed in the Introduction and Discussion, particularly because A-B pairs were overtrained on Day 1. I think that the manuscript would benefit from more explicit characterization of why the adopted paradigm speaks to a suppression account of a B vs. an integration account of a B in memory updating, and how their data adjudicate between these two accounts (e.g., MPFC is often shown as a schema/generalization area).

We thank the reviewer for this very thoughtful comment. We completely agree with this reviewer that our paradigm is akin to the broad area of memory changes, including integration, inhibition, and/or differentiation. The Introduction has now been significantly reframed. In particular, we have now introduced alternative hypotheses in the Introduction and provided results to adjudicate between these accounts. As stated in our response to the first reviewer’s comment, our new behavioral results and fMRI data provide support to the integration and differentiation hypothesis.

2) Analytic technique:a) I was not convinced that the retrieval practice vs. re-study Day 2 design and analyses provided clear support for the idea that B is suppressed and C is prioritized in retrieval practice vs. re-study during memory updating. In fact, the authors do not find evidence of this claim in their follow-up behavioral study designed to address the issue; they have to split data to show that in some subjects you see this pattern but in others you don't. Because the evidence stemming from this follow-up is not clear, the mechanism of retrieval practice in memory updating (suppression vs. integration) is not clear.

Thanks for this question. As stated in our response to the first reviewer’s comments, we have done an additional experiment (new Experiment 2) to examine the suppression vs. integration vs. differentiation hypotheses. We found the A-B memory was very strong and was not suppressed by retrieval practice, despite the consistent retrieval practice effect on A-C memory. These new results, together with the data from Experiment 3 (Experiment 2 in original submission), clearly support the differentiation hypothesis. That is, retrieval practice might help to form non-overlapping representations that also reduce competition. Additional joint analysis on how subjects maintained both A-B and A-C memories further supported the differentiation hypothesis, as we found that compared to Restudy, RetPrac had more trials where subjects correctly remembered both A-B and A-C memories, but fewer trials where subjects answered the old B memory on both tests. Finally, the new analysis of fMRI data suggested that the MPFC showed strong and comparable competitor reactivation for both correct and incorrect responses under the RetPrac condition, further supporting the integration and differentiation account.

b) The critical results of the study are subsection “Retrieval practice enhanced target reactivation and competitor suppression” of the manuscript. I was confused as to why the authors have one classifier analysis for final test performance (Day 3) that doesn't account for retrieval practice/re-study (Day 2), and then another classifier analysis for retrieval practice/re-study (Day 2) that doesn't account for final test performance (Day 3). Why was behavioral performance but not the Day 2 manipulation used in the first analysis, and why was the Day 2 manipulation but not behavioral performance used in the second analysis? I think an analysis that uses both these assays would get at the authors' question most directly.

Following this reviewer’s insightful suggestion, we have conducted new three-way ANOVA to simultaneously examine the effects of behavioral performance and updating method on the target and competitor evidence. The new analysis revealed interesting results that supported the retrieval-induced integration and differentiation account in the MPFC. Meanwhile, we also found that RetPrac reduced the competitor evidence in VTC. Finally, the AG tracked the behavioral performance and RetPrac did not have an additional effect on AG representation. These results thus showed clearly dissociated effects of RetPrac in different brain regions. We have reported the results from the new analysis in the revised manuscript.

c) How many trials were used in the fMRI analyses per condition?

There were 72 trials in each condition.

d) Greater motivation for the ROI selection parameters for MVPA should be given.

We have added a more detailed description of the motivation for the ROI selection, which reads:

“We focused our MVPA on the ventral temporal cortex (VTC), medial prefrontal cortex (MPFC), angular gyrus (AG), and hippocampus (HPC), which overlap with the core recollection network (Rugg and Vilberg, 2013) and have consistently shown neural reinstatement effect during memory retrieval (Kuhl and Chun, 2014; Wimber et al., 2015; Xiao et al., 2017).”

e) Is it possible that shifts in decision criteria would be observed for those trials with retrieval practice on Day 2 vs. those trials with re-study on Day 2? Can the authors rule out a decision criteria account of behavioral findings on Day 3 final test?

Thanks for this comment. In the final test on Day 3, all trials from different conditions were mixed together, thus we believe a shift of decision criteria would be unlikely. We found that the response of the other category was matched between the two conditions.

3) Conclusions drawn from the data:a) As noted, I did not think that the critical finding about retrieval practice during memory updating was supported by the data; the authors' own follow-up study did not provide strong behavioral support for this claim. Thus, it is hard to reconcile the lack of a clear mechanism with the fMRI side of the manuscript.

As stated above, with an additional behavioral experiment and a new analysis of the fMRI data, our results support the memory differentiation account. Memory differentiation helps to reduce the intrusion in the A-C memory test.

b) The lack of neuroimaging effects for AG and Hipp was striking, particularly given recent work finding reactivation effects in parietal cortex (Jonker, Ranganath, 2019, PNAS; Lee, Kuhl, 2019, Cerebral Cortex). These papers should be cited in text and discussion should be given as to the discrepancies.

Thanks for this comment and the references. In our new analysis that simultaneously examined the effects of updating method and performance, we found that AG was tracking the behavioral performance and updating method has no additional effect. This pattern was quite consistent with previous studies showing that AG reactivation was aligned with behavioral performance. For the hippocampus, we did not find strong representation of category information in this region, which was also consistent with existing literature. We have added more discussion on these findings, by referring to the literatures suggested by this reviewer, which reads:

“Several major reasons might account for the chance-level decoding of memory information in the hippocampus during retrieval. […] Future studies should use an optimized design and item-level analysis to further elucidate the hippocampus’s role in memory updating”

Reviewer #3:[…]1) This is a very interesting and fruitful set of results, but the framing does not seem to fit with the work that is reported. The logic of the paper is that neural changes in response to repeated retrievals reflect memory updating and suppression of old memory, but a priori this did not seem like an obvious prediction. It is possible that any differences (both in behavioral performance and neural activity) between the test and the restudy condition simply reflect superior learning during testing. In other words, without a no interference or weaker interference condition, it is hard to conclude that brain activity during retrieval practice supports memory updating/suppression.

We thank this reviewer for raising this important question. The testing effect is complicated, which could reflect the strong A-C learning, A-B inhibition, and/or memory differentiation. We completely agree with this reviewer that testing would lead to superior learning of new memory, which was found in our behavioral and fMRI data. In fact, we believe that if no interference or only weak interference is introduced, one major benefit of retrieval practice would be the superior learning of new memories. This effect has been consistently shown in the literature.

Building upon this literature, the current study aimed to examine whether and how retrieval practice could facilitate memory updating when there was strong interference. In particular, we examined whether there were additional mechanisms that allow retrieval practice to facilitate memory updating, such as competitor suppression and/or memory integration and differentiation. Our results showed that retrieval practice could facilitate target representation in the MPFC, and reduce competitor representation in the VTC. Furthermore, we found that retrieval practice could integrate and differentiate target and competitor evidence in the MPFC. These results suggest that retrieval practice engages multiple, region-specific mechanisms to facilitate memory updating. In this revision, we have clearly emphasized these points throughout the manuscript.

2) From the Introduction, it is unclear how this study is different from prior studies examining neural mechanisms underlying retrieval induced forgetting using a similar paradigm, e.g. Wimber et al., 2015.

As stated above, we have now significantly reframed the Introduction to emphasize the multiple mechanisms of retrieval practice on memory updating.

3) The authors capture many previous findings about activity in mPFC and LPFC, however, the hypothesis they form following this literature is too vague to be clearly falsifiable or to adjudicate between potentially contradictory findings. For instance, rather than reflecting suppression, stronger reactivation of competing memory during retrieval practice has also been associated with retrieval induce facilitation (e.g. Jonker et al., 2018). Moreover, the authors examine brain activity in 4 ROIs but did not present a rationale for including AG, IFG, VTC.

As correctly pointed out by this reviewer, existing literature found that the reactivated memory could be integrated, strengthened or differentiated, depending on mnemonic goals and the characteristics of the reactivated memory and(Tambini Davachi, 2019). We have made this idea clearer in the Introduction, and added more discussion on these different results. We have also added more rationales for including AG and VTC, and made clearer the prediction regarding the roles of MPFC and LPFC in retrieval practice and memory updating.

4) It is laudable the authors report a follow-up behavioral experiment examining the relationship between memory for new memory vs. old memory. However, the negative correlation could be driven by output interference, especially given that subjects recalled A-C pairs first. It is likely that better A-C recall produced stronger output interference to A-B pairs.

This is a very good point. As stated in our response the other two reviewers’ comments, we have done an additional behavioral experiment to examine the effect of retrieval practice on A-B memory, balancing the order of the memory tests (recall A-B or A-C first). Our results support the differentiation hypothesis.

5) It is unclear in the analyses of retrieval practice reactivation, whether the authors included all test trials or only correct test trials. If all trials were included, all the results reported in these sections would not be surprising because subjects wrongly recalled a large portion old targets during retrieval practice. This also explains why restudy trials showed larger reactivation of new memory because new information was always directly presented.

Thanks for this comment. In the analysis of competitor reactivation, we included all trials. The reviewer was correct that subjects recalled a large number of old targets, which would have contributed to the reactivation of competitor evidence. The rationale for including all trials was to show that subjects made many mistakes and reactivated the old target memory under the retrieval practice condition. Whereas under the restudy condition, the competitor evidence was not strongly reactivated because the new information was directly presented.

It should be noted that the strong competitor reactivation was not simply a result of behavioral intrusion of old memory. For example, we found the competitor evidence was stronger as compared to other evidence (i.e., baseline), although subjects made comparable competitor and other responses.

Following this reviewer’s comment, we did an additional analysis to include only correct trials, which still revealed stronger competitor reactivation under the RetPrac condition than under the restudy condition (all ps <.017). We have included the new result in the supplementary materials (Figure 4—figure supplement 1).

6) A number of ad hoc hypotheses are given for results that are inconsistent with the prediction. For example, the authors claim that the null results for correct or incorrect trials are due to a small number of trials. However, there should be at least 70 correct trials in the test condition and 50 correct trials in the restudy condition. Moreover, decreased update in the restudy condition is explained by the repetition suppression effect; null results of correlations between competitor reactivation and behavioral performance are explained by the claim that behavioral measure is not sensitive; chance level classification performance in the hippocampus is thought to reflect "technical limitations". These ad hoc explanations of conflicting results, which lack justification, suggest strong confirmation bias.

Thanks for these comments.

“For example, the authors claim that the null results for correct or incorrect trials are due to a small number of trials. However, there should be at least 70 correct trials in the test condition and 50 correct trials in the restudy condition.” We have done a new analysis to simultaneously examine the effects of updating method and behavioral performance on target and competitor evidence. We have updated the results in this revision.

“Moreover, decreased update in the restudy condition is explained by the repetition suppression effect”: We have removed the results from this analysis due to differences in task structure between RetPrac and Restudy.

“null results of correlations between competitor reactivation and behavioral performance are explained by the claim that behavioral measure is not sensitive”: This result has been removed from this revision.

“chance level classification performance in the hippocampus is thought to reflect "technical limitations": For the hippocampus, we found that during the localizer task, the hippocampus showed above-chance classification, but the accuracy was lower than other regions. Furthermore, we found that the hippocampal classifier could not predict subjects’ response during the final test. This could be due to several reasons. We have added more discussion on this, which reads:

“Several major reasons might account for the chance-level decoding of memory information in the hippocampus during retrieval. […]Future studies should use an optimized design and item-level analysis to further elucidate the hippocampus’s role in memory updating”

[Editors’ note: what follows is the authors’ response to the second round of review.]

Revisions:1) The authors analyzed the behavioral data with a "test order (A-C first, A-B first) by memory test type (Recall A-C, Recall A-B) by update method (RetPrac, Restudy) by response type (Target, Competitor, Other) four-way mixed design ANOVA". Either the choice or the description of the analysis is incorrect. Given that the performance of A-C test and A-B test, absolute ratios of Target, Competitor and Others are not comparable, response type and memory test type should not be used as independent variables (factors). Rather, separate ANOVAs should be conducted, with appropriate multiple comparison correction, to examine target ratio and competitor ratio in each test.2) Given the above reason, the authors cannot rule out the potential order confounding by just showing there was no significant effect of test order or interaction effect with test order in this four-way ANOVA. Rather, the authors need to directly compare A-B performance between when A-B was tested first vs when A-B was tested after A-C. However, even the correct analysis was done, non-significant difference between different orders cannot rule out the confounding of order either. A stronger test would be only examining A-B performance with subjects start with A-B test, and vice versa for A-C test.

We thank the reviewers for pointing out this important issue. Following the reviewers’ suggestion, we have done six test order (A-C first, A-B first) by update method (RetPrac, Restudy) two-way ANOVAs to examine the test order effect, separately for A-C memory test and A-B memory test, and each response type (Target, Competitor, Other). The results reveal no significant main effect of test order (ps >.43, except a trend of significant effect in Other responses during A-B test, p = .07, FDR corrected). Importantly, there was no update method by test order interaction in any of the model (all ps >.54, FDR corrected). These results indicated none or neglectable effect of test order in our data.

We agree with the reviewers that a stronger test of the difference between A-B and A-C memory performance should only use half the data, since we would predict that test order could affect the A-B and A-C memory performance. More specifically, A-B memory would be worse when A-C memory was tested first than when A-B memory was tested first, and vice versa for A-C memory. In this case, even though there was no statistically significant test order effect, it might still confound the data.

Nevertheless, the current study was interested in the effect of updating method. For several reasons, we think it might be better to use all data in our analysis. First, we should not expect strong test order by updating method interactions, and our data confirmed this, and none of the p value is even close to significant level. Second, we found very consistent results in both subjects group (A-C test first and A-B test first). For example, we found that both groups showed effect of updating method on A-C memory during A-C test (Author response image 2). However, it would be tedious to separately report these results when there was not even a trend of interaction. Third, more subjects could generate more reliable results and enable us to detect more subtle effect in our additional joint analysis. We have clarified this issue in this revision, which reads:

“For both A-C memory and A-B test and each response type (Target, Competitor, Other), test order (A-C first, A-B first) by update method (RetPrac, Restudy) two-way ANOVA revealed neither significant main effect of test order (ps >.43, except a trend of significant effect in Other responses during A-B test, p = .07, FDR corrected), nor test order by update method interaction (ps >.54, FDR corrected) (Supplementary file 1). Given our focus on the effect of updating method and the lack of updating method by test order interaction, we thus combined data from both groups in the following analyses to increase the statistical power.”

**Author response image 2. respfig2:**